# Periodic dynamic induction control of wind farms: proving the potential in simulations and wind tunnel experiments

Joeri Frederik[a], Robin Weber[b], Stefano Cacciola[c], Filippo Campagnolo[b], Alessandro Croce[c], Carlo Bottasso[b], and Jan-Willem van Wingerden[a]

[a]Delft Center of Systems and Control, Faculty of Mechanical, Maritime and Materials Engineering (3mE), TU Delft. Mekelweg 2, Delft, The Netherlands
[b]Wind Energy Institute, Technische Universität München, Garching bei München, D-85748 Germany
[c]Dipartimento di Scienze e Tecnologie Aerospaziali, Politecnico di Milano, Milano, I-20156 Italy

**Correspondence:** Joeri Frederik (J.A.Frederik@TUDelft.nl)

**Abstract.** As wind turbines in a wind farm interact with each other, a control problem arises that has been extensively studied in literature: how can we optimize the power production of a wind farm as a whole. A traditional approach to this problem is called induction control, in which the induction factor set-point, i.e., the in-wake wind speed deficit, of a turbine is lowered such that downstream turbines can increase their power capture. In recent simulation studies, an alternative approach, where the induction factor is varied over time, has shown promising results. In this paper, the potential of this Dynamic Induction Control (DIC) approach is further investigated. Only periodic variations, where the input is a sinusoid, are studied. A proof of concept for this periodic DIC approach will be given by execution of scaled wind tunnel experiments, showing for the first time that this approach can yield power gains in real-world wind farms. Furthermore, the effects on the Damage Equivalent Loads (DEL) of the turbine are evaluated in a simulation environment. These indicate that the increase in DEL on the excited turbine is limited.

## 1 Introduction

The interaction between wind turbines in a wind farm through their wake is a field of research as old as wind farms themselves. The wake of a turbine has a wind field with a lower velocity and a higher Turbulence Intensity (TI), resulting in a lower power production and higher relative loads for downstream turbines. To exploit this interaction between turbines, induction control (sometimes called "derating"), with induction the in-wake speed deficit, has been a popular research topic in recent years. The concept of this control approach is schematically shown in Figure 1a. Despite initial promising results (Marden et al., 2013; Gebraad et al., 2013), recent studies indicate that the power gain that can be achieved with steady-state induction control is limited to non-existing (Campagnolo et al., 2016a; Nilsson et al., 2015; Annoni et al., 2016).

An alternative approach, first mentioned in (Westergaard, 2013), is to actively manipulate wake recovery. Recent simulation studies (Goit and Meyers, 2015; Munters and Meyers, 2017) have shown that so-called Dynamic Induction Control (DIC) improves the power production in small to medium-sized wind farms. This approach, where the induction factor is varied over time, generates a turbulent wind flow that enables enhanced wake recovery. Consequently, downstream turbines will

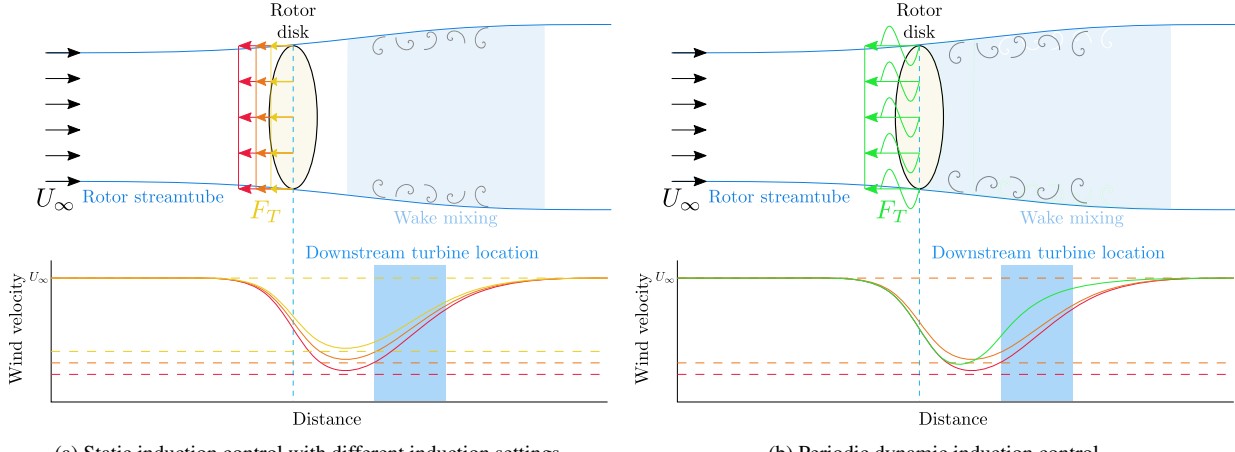

| (a) Static induction control with different induction settings. | (b) Periodic dynamic induction control. |

**Figure 1.** A schematic representation of a wind turbine in a flow field, showing the working principles of static (a) and dynamic induction control (b). On the top, the turbine is simplified as a rotor disk, and its streamtube - the area where the wind speed is affected by the turbine settings - is depicted. The force $F_T$ exerted on the wind is shown for different induction settings, where red depicts greedy control, orange and yellow arbitrary static derating settings, and green periodic DIC. The bottom figures show the corresponding wind velocity profiles, with respect to inflow velocity $U_\infty$, as a function of the distance from the turbine. The area highlighted in blue is where a downstream turbine is typically located.

compensate for the power loss of the upstream turbine, leading to a higher overall power production of the wind farm. The optimal dynamic control inputs are found using a computationally expensive adjoint-based Model Predictive Control (MPC) approach.

In Munters and Meyers (2018), a simpler approach is suggested: the induction variation is limited to a sinusoidal signal implemented on an actuator disk. This approach is here dubbed "periodic DIC". A grid search with different amplitudes and frequencies is performed to find the optimal dynamic signal in a high-fidelity simulation environment. The effect of this approach on the streamtube and downstream wind velocity is shown in Figure 1b. It should be noted that the applied excitation is very low-frequent. An optimal Strouhal number $St = 0.25$ is found, which corresponds to a period of approximately 56 seconds for an NREL 5 MW turbine (Jonkman et al., 2009).

However, no experiments have yet been executed that validate this approach on actual, either scaled or full-sized, wind turbines. Furthermore, the effects of DIC on the loads of the turbines are yet to be evaluated. This paper aims to bridge this knowledge gap by executing a thorough evaluation of DIC both in simulation environments and in wind tunnel experiments. The effects of DIC on the loads on turbine level are evaluated using the aeroelastic tool CP-LAMBDA (Bottasso and Croce, 2009–2018; Bottasso et al., 2006). For the wind tunnel experiments, the Atmospheric Boundary Layer (ABL) wind tunnel of

the Politecnico di Milano (Polimi) is used (Bottasso et al., 2014). Three `G1` models, which have a rotor diameter of 1.1 m and are developed by the Technical University of Munich (TUM) (Campagnolo et al., 2016a, b, c) will be used as turbine models.

To verify the validity of the periodic dynamic induction approach for fast wake recovery in a wind farm, a number of wind tunnel experiments in both low and high Turbulence Intensity (TI) conditions are executed. All experiments are executed at a below-rated wind speed, i.e., in operating region II. The effect of varying the amplitude and frequency of the signals is studied, and the performance of this approach is compared with other state-of-the-art wind farm control strategies. A positive result in these experiments would be an important step towards proving the validity of this approach in real wind farms.

The structure of this paper will be as follows: in Section 2, the DIC strategy will be explained. Sections 3 and 4 will elaborate on the simulation environment and the experimental setup, respectively. In Section 5, the simulation results will be presented, followed by the experimental results obtained in the wind tunnel in Section 6. Finally, the conclusions will be drawn in Section 7.

## 2   Control Strategy

In this section, the strategy behind dynamic induction control will be discussed shortly. As mentioned in the introduction, the approach presented in Munters and Meyers (2018) is used as a basis for this paper: the thrust force of the upstream wind turbine is excited to induce wake mixing, in order for downstream turbines to increase their power capture. It is shown that the amplitude and frequency of a sinusoid determine the overall power production. The optimum found in here is a Strouhal number of $St = 0.25$, with an amplitude of the disk-based thrust coefficient $C'_T = 1.5$. The Strouhal number is defined as $St = fD/U_\infty$ for a given frequency $f$, rotor diameter $D$ and inflow velocity $U_\infty$, while $C'_T = 4a/(1-a)$, with $a$ the axial induction factor (Goit and Meyers, 2015). This disk-based thrust coefficient relates to the thrust coefficient $C_T$ as $C_T = C'_T(1-a)^2$. For the G1 models and an inflow velocity of $5.65\,\text{m/s}$, this Strouhal number would result in an excitation frequency of approximately $1.3\,\text{Hz}$.

However, there are some fundamental differences between Munters and Meyers (2018) and the work presented here, which are summarized in Table 1. Due to the size of the wind tunnel (see Section 4), a 3-turbine wind farm is the deepest possible array configuration. The amplitude and frequency ranges were slightly reduced due to time constraints. Finally, to allow for practical implementation on a turbine model, the collective pitch angle $\beta$ of the upstream model was excited periodically. This results in a slightly different thrust signal, as shown in Figure 2, but simulations show that the difference in output for these input signals is limited.

Since the internal torque controller of the G1 model is also active, the amplitudes and offsets of the pitch signals are tuned manually such that the resulting thrust coefficient matches the desired thrust coefficient in amplitude and frequency. To achieve this, the thrust force on the turbine is measured, which, together with knowledge about the wind conditions, is used to calculate the thrust coefficient over time.

Finally, a comparison will be made with wind farm control approaches that have already been investigated more extensively in literature: static induction control (also called derating control) and yaw control (also called wake redirection control). The optimal control settings are found using the static FLORIS model (Annoni et al., 2018). This parametric model is calibrated with wind tunnel measurements, as described in Schreiber et al. (2017). The control settings are then implemented on the same

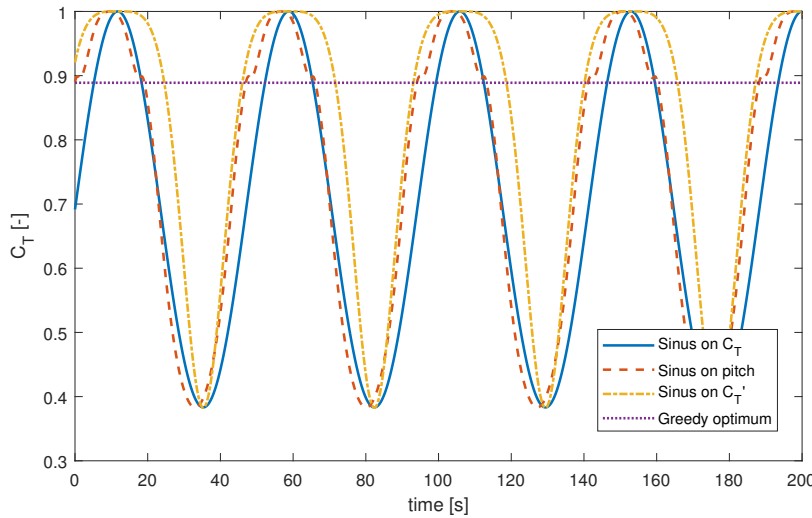

**Figure 2.** Values of $C_T$ for different types of input signals, created using a look-up table of the G1 turbine model. The thrust coefficient is shown for three different sinusoidal excitations: on $C_T$, on $C_T'$ and on the collective pitch angle $\beta$, tuned such that the amplitude of $C_T'$ is 1.5. The dashed line shows the steady-state optimal $C_T$.

wind farm set-up in the wind tunnel such that a fair comparison can be made. In Section 6, the results of these experiments will be evaluated.

## 3 Simulation environment

In order to evaluate the effect of DIC on turbine level, the aeroelastic tool `Cp-Lambda` (Code for Performance, Loads, Aeroe-
5 lasticity by Multi-Body Dynamics Analysis) (Bottasso and Croce, 2009–2018; Bottasso et al., 2006) has been used. This soft-
ware is an aeroelastic code based on finite element multibody formulation, which implements a geometrically exact non-linear
beam formulation (Bauchau, 2011) to model flexible elements such as blade, tower, shaft and drive train. The generator-drive

**Table 1.** Differences between the approach in Munters and Meyers (2018) and both the simulations and wind tunnel experiments presented in this paper. Note that the pitch amplitude $\beta = 2°$ used in the simulations leads to a amplitude of approximately $C_T' = 1.5$.

|  | **Munters et. al.** | **Simulations** | **Experiments** |
|---|---|---|---|
| **Layout** | 4 turbines in a row | Single turbine | 3 turbines in a row |
| **Environment** | LES code | Aero-elastic code | Wind tunnel experiments |
| **Control input** | Sinusoid on $C_T'$ | Sinusoid on $\beta$ | Sinusoid on $\beta$ |
| **Amplitude of excitation** | $C_T'$ of 0.5, 1, 1.5 and 2 | $\beta = 2°$ | $C_T'$ of 1, 1.5 and 2 |
| **Strouhal number $St$ of excitation [-]** | Between 0.05 and 0.6 | Between 0.3 and 0.5 | Between 0.09 and 0.41 |

train model can include speed-dependent mechanical losses. The rotor aerodynamics are modelled via blade element momentum (BEM) theory or a dynamic inflow model, and may consider corrections related to hub- and tip-losses, tower shadow, unsteadiness and dynamic stall, whereas lifting lines can be attached to both tower and nacelle to model the related aerodynamic loads.

For the fatigue analysis, the model of the NREL 5 MW reference wind turbine (Jonkman et al., 2009) was considered. This reference 5 MW wind turbine has a rotor diameter of 126 m and a rated wind speed of 11.4 m/s. Each blade is discretized with 30 cubic finite elements, the tower with 20 cubic elements. Additionally, pitch and torque actuators are modeled respectively as second and first order systems and the model is completed by a standard PID controller (Jonkman et al., 2009). Finally, 10-minute wind time histories of turbulence class "A", according to DLC 1.1 of IEC 61400-1 Ed.3. (2004), generated by the

software `TurbSim` (Jonkman and Buhl, 2006), were given as input to the aeroelastic solver.

## 4   Experimental Setup

The experimental results presented in this paper were gathered by performing dedicated tests within the wind tunnel of the Politecnico di Milano (Polimi), which is a closed-return configuration facility arranged in a vertical layout and equipped with two test rooms. A detailed description of the facility can be found in (Bottasso et al., 2014). The tests were performed within

the boundary layer test section, which has been conceived for civil, environmental and wind energy applications. This section has a large cross-sectional area of $13.84 \times 3.84$ m, which allows for low blockage effects even with several relatively large turbine models installed within the test section.

    Roughness elements located on the floor and turbulence generators placed at the chamber inlet are commonly used to mimic to scale the atmospheric boundary layer in terms of vertical shear and turbulence spectrum. During the experiments

described later on, two boundary layer configurations were used: one generating low turbulent (low-TI) and one generating highly turbulent (high-TI) flow conditions. These conditions roughly correspond to off- and onshore operation respectively. The flow characteristics are shown in Figure 3 together with the extension of the model's rotor disk along the vertical axis. The coefficients of the vertical-shear exponential law, shown in the same picture, that best fit the experimental data are 0.144 and 0.214 for the Low-TI and High-TI cases respectively.

### 4.1   Wind turbine models

Three `G1` wind turbine models developed at TUM were used to perform the experiments reported in this paper. This model type was widely employed and described in detail in previous research (Campagnolo et al., 2016a, b, c) and is shown within the boundary layer test section of the Polimi wind tunnel in Figure 4. The setup of the turbines in the tunnel is shown in Figure 5.

    With a rotor diameter of $D = 1.1$ m and a rated rotor speed of $850$ rpm, the model was designed to have a realistic energy

conversion process and wake behavior: it exhibits a power coefficient $C_\mathrm{P} \approx 0.41$ and a thrust coefficient $C_\mathrm{T} \approx 0.81$ for a tip speed ratio $\lambda \approx 8.2$ and a blade pitch $\beta \approx 0.4°$.

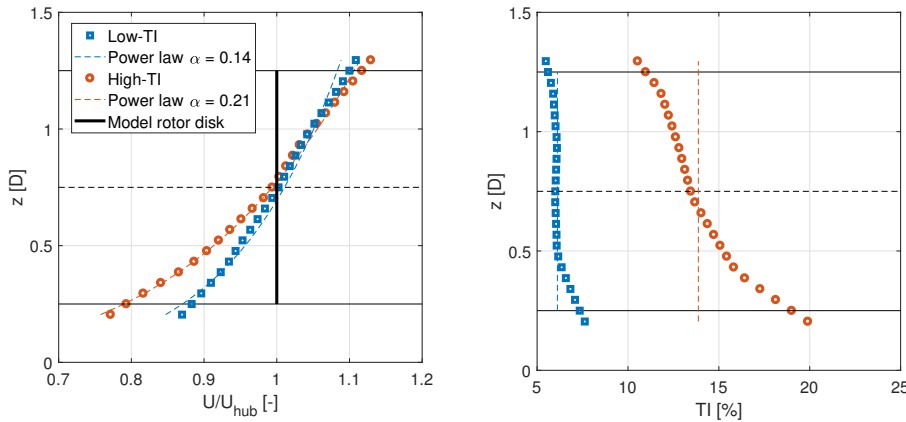

**Figure 3.** Vertical wind speed profile (a) and turbulence intensity (b) as a function of height above the tunnel floor, for low (low-TI) and high (High-TI) turbulence experiments.

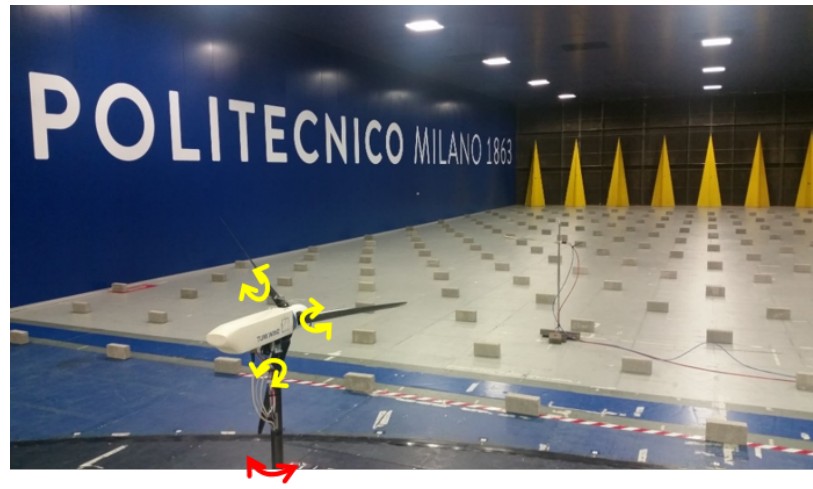

**Figure 4.** A G1 scaled wind turbine model within the wind tunnel of the Politecnico di Milano. The yellow and red arrows show the pitch and yaw control possibilities respectively. The yellow spires and bricks in front of the model create the high-TI flow conditions.

The turbine is actively controlled with individual pitch, torque and yaw actuators and features comprehensive on-board sensorization. Three individual pitch actuators and connected positioning controllers allow for an overall accuracy of the pitch system of 0.1 degrees for each blade and the ability to oscillate the blade pitch with an amplitude of 5 degrees at 15 Hz around any desired pitch angle. Strain gauges are installed on the shaft to measure bending and aerodynamic torsional loads, as well as at the tower foot to measure fore-aft and side-side bending moments. A pitot tube, placed three rotor diameters upstream of the first turbine model, provides measurements of the undisturbed wind speed at hub height. Finally, air pressure, temperature

and humidity transducers allow for measurements of the air density within the test section. The measurements of these sensors are used to determine the performance of the turbine models. The thrust coefficient is obtained using measurements of the pitot tube wind speed measurement and fore-aft bending moment, while correcting for the effects of the tower and nacelle drag.

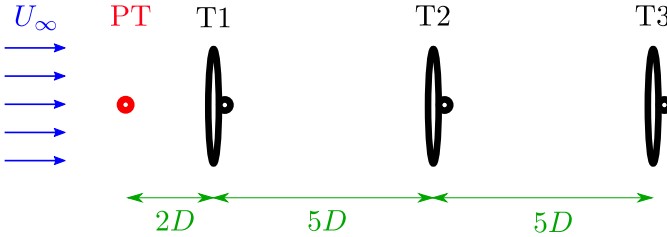

**Figure 5.** A schematic top view of the wind farm setup in the wind tunnel. The pitot tube (PT), which measures the inflow velocity, is located 2 rotor diameters $D$ in front of Turbine 1 (T1). The spacing between the turbines is $5D$ and the wind flows from left to right.

### 4.2 Control system

For each wind turbine model, control algorithms are implemented on a real-time modular Bachmann M1 system. Demanded values (e.g. pitch angle or yaw angle references) are then sent to the actuators, where the low level control is performed. Torque signals, shaft bending moments and rotor azimuth position are recorded with a sampling rate of 2.5 kHz, while all other measurements are acquired with a sampling rate of 250 Hz. A standard power controller is implemented on each M1 system based on Bossanyi (2000), with two distinct control regions. Below rated wind speed, blade pitch angles are kept constant, while the generator torque reference follows a function of the rotor speed with the goal of maximizing the energy extraction. Above rated wind speed, the generator torque is kept constant and a proportional-integral (PI) controller adjusts the collective pitch of the blades in order to keep the generated power at the desired level. All experiments presented in this work are performed below rated wind speed.

For the tests performed within the research described in this paper, the standard power controller was augmented in order to enable the rotor thrust coefficients following a specific sine wave function. However, there is not a unique way of achieving this goal, since a specific thrust coefficient $C_T(\lambda, \beta)$ can be obtained by operating at different combinations of tip-speed-ratio $\lambda$ and blade pitch $\beta$. In turn, the tip speed ratio can be varied either by changing the reference followed by the generator torque or changing the blade pitch. In this paper, a strategy that only changes the blade collective pitch is adopted. The implementation of this strategy simply requires changing the collective fine pitch at which the model blades are set when the machine operates in partial load conditions (region II). The fine pitch was tuned experimentally, by means of a trial and error procedure conducted with a stand-alone model, to achieving the desired mean $\bar{C}_T$ and amplitude $A$ as reported in Table 2. The effects of these control actions in terms of impacts on the power output of the 3-turbine wind farm will be discussed in Section 6.

**Table 2.** Average $\bar{C}_T$ and amplitude $A_{C_T}$ of the three different thrust coefficient oscillations whose results are discussed in Section 6, as well as the mean pitch angle average $\bar{\beta}$ and amplitude $A_\beta$ used to achieve these signals. Note that, as explained in Section 2, these collective pitch settings are not identical for different frequencies. Instead, they are tuned such that the mean and amplitude of $C_T$ as given below are followed as accurately as possible.

| Amplitude $C_T'$ | $\bar{C}_T$ [-] | $A_{C_T}$ [-] | $\bar{\beta}$ [deg] | $A_\beta$ [deg] |
|---|---|---|---|---|
| $A = 1$ | 0.8 | 0.17 | 0.7 | 1.7 |
| $A = 1.5$ | 0.7 | 0.3 | 1.8 | 2.8 |
| $A = 2$ | 0.5 | 0.5 | 4 | 5 |

## 5   Simulation Results

To evaluate the effects of DIC on the loads of the excited turbine, a full set of aeroelastic turbulent simulations (DLC 1.1) has been executed. These analyses have been conducted on the NREL 5 MW wind turbine with the main goal of quantifying the effect of this DIC on the fatigue loads. The analysis focuses mainly on the main wind turbine sub-components, such as the

blade root flap- and edge-wise loads, as well as the tower base fore-aft bending and hub torsional moments.

DIC was assumed to be activated for wind speeds between 3 and 25 m/s, to cover the totality of regions I-1/2, II, II-1/2 and III. Notice that 25 m/s seems a rather high speed, considering the fact that so far, the effectiveness of DIC has only been evaluated in region II. In region III, the lower rotor inductions (i.e a lower in-wake speed deficit) may guarantee, together with the high inflow velocity, the full power region for the downwind rotor(s). Nevertheless, in the 10-minute simulation, the high

turbulence intensity (class "A") causes a relatively long period where the mean wind speed is below the rated one and hence DIC may have an important effect on the wake. From this point of view, extending the authority of DIC up to 25 m/s is to be regarded as a conservative choice. For clarity, the rated wind speed of 11.4 m/s will be shown in the figures showing the DELs at different mean wind speeds.

Strouhal numbers of $St = [0.3, 0.4, 0.5]$ and a pitch amplitude $\beta_{DIC} = 2°$ were used in the aeroelastic simulations of the

5 MW turbine. Considering the diameter of this wind turbine model (126 m), the frequency of DIC $f_{DIC}$ is between $6.94 \cdot 10^{-3}$ Hz at 3 m/s (and $St = 0.4$) and $5.95 \cdot 10^{-2}$ Hz at 15 m/s (and $St = 0.5$), which correspond to a period equal to between 105 and 16.8 s respectively.

Due to the relatively low excitation frequency, the baseline turbine control is able to trim the machine without a significant additional effort or detrimental performance. Moreover, a coalescence between the DIC input frequency and turbine vibratory

modes is not to be expected, at least for on-shore or off-shore turbines installed on rigid foundations.

Figure 6 shows an example of the time response of the machine with and without DIC. These simulations have been performed with a Normal Turbulence Model (NTM) of class-A wind (IEC 61400-1 Ed.3., 2004) with a mean hub wind speed of 9 m/s, generated with `TurbSim` (Jonkman and Buhl, 2006). In these conditions, the wind turbine baseline control switches between region II, II-1/2 and III. The figure shows the baseline condition, i.e., the one without the DIC controller, and two

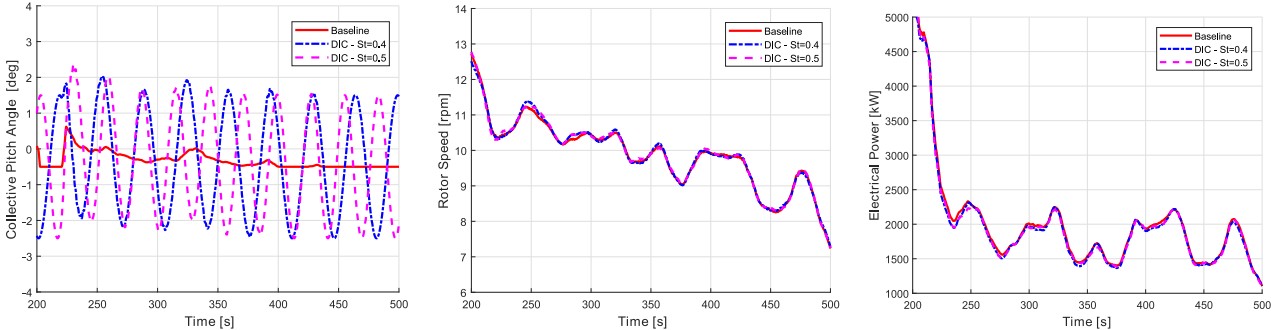

**Figure 6.** Comparison of pitch activity (left), rotor speed (middel) and power (right) between baseline (solid red) and DIC controlled with $St = 0.4$ (dash-dotted blue) and $St = 0.5$ (dashed magenta) turbine for NTM class "A" at 9 m/s.

simulations with Strouhal number $St = 0.4$ and $St = 0.5$. The plot on the left refers to the pitch activity, the plot in the middle to the rotor speed and the plot on the right to the power. The collective pitch angle time histories show the DIC activity superimposed to the trim-pitch. As can be seen, the rotor speed and power production with DIC active behave very similar to that of the baseline case (solid lines), showing that the addition of the periodic pitch motion is not detrimental in terms of trimmer performance.

Figure 7 shows the power spectral density (PSD) of the rotor speed (left) and blade root flapwise bending moment with a NTM at 15 m/s, again for the baseline case (solid-red) and for DIC with Strouhal numbers $St = 0.4$ and $St = 0.5$. Both figures show a new frequency corresponding to the DIC excitation. This peak is far from the other aeroelastic frequencies of the wind turbine (the first being the tower fore-aft at $f = 0.31 Hz$), but may have an important role on the fatigue loads.

From the 10-minute simulations computed according to DLC 1.1 of IEC 61400-1 Ed.3. (2004), the stochastic time histories of the wind turbine loads are converted into simplified Damage Equivalent Loads (DELs) through a rainflow analysis and depicted in Figures 8 and 9 as a function of the mean wind speed. These figures show that DELs computed for the baseline case are almost always lower compared to when DIC is active, as would be expected based on Figure 7. For each mean wind speed, the DIC frequencies correspond to Strouhal numbers $0.4$ and $0.5$. Even though DIC is only effective at lower wind speeds, it is assumed active in the entire region III. As can be seen, the tower base fore-aft bending moment and the blade root flapwise are affected the most by this controller. As expected, the blade edge-wise bending moment is only slightly affected, since the DEL in edge-wise direction is mainly driven by gravity.

In order to have a more comprehensive indication about the impact of DIC on fatigue loads, one can consider the Weibull-weighted DELs, i.e., the DELs weighted throughout the probability distribution of the wind as expressed by the Weibull distribution $p_w(V)$

$$p_w(V) = k \frac{V^{(k-1)}}{C^k} \mathrm{e}^{-\left(\frac{V}{C}\right)^k}, \tag{1}$$

where $k$ is the shape parameter and $C = 2V_{\mathrm{av}}/\sqrt{\pi}$ the scale factor and $V_{\mathrm{av}}$ the average wind speed.

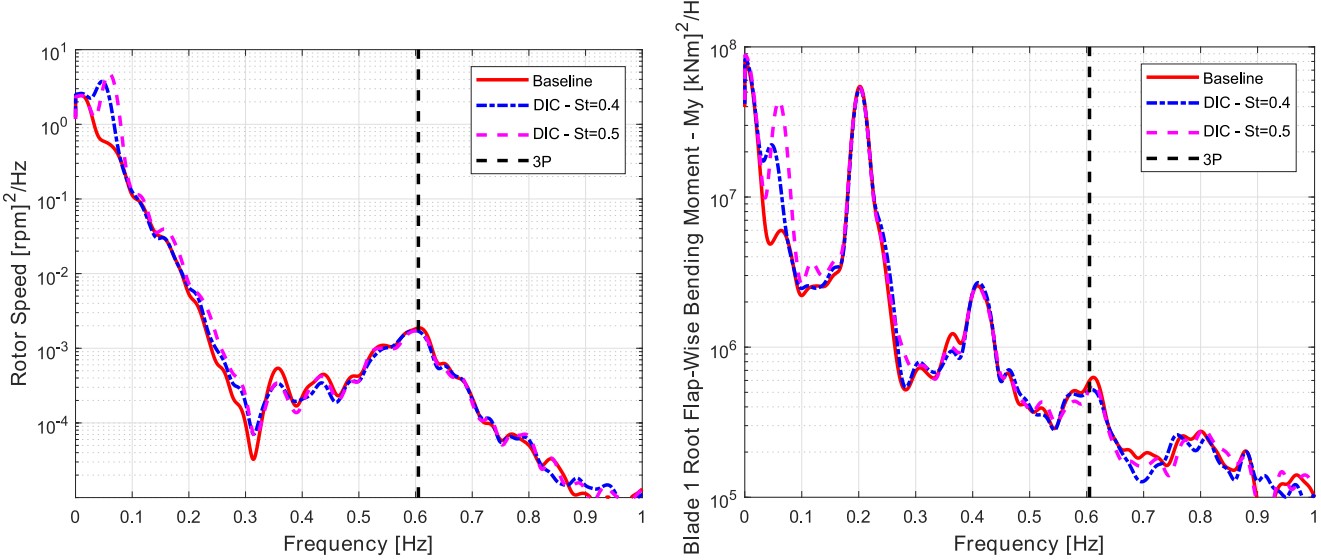

**Figure 7.** PSD comparison of the rotor speed (left) and blade root flap-wise bending moment (right) between baseline (solid red) and DIC controlled with $St = 0.4$ (dash-dotted blue) and $St = 0.5$ (dashed magenta) turbine for NTM class "A" at 15 m/s.

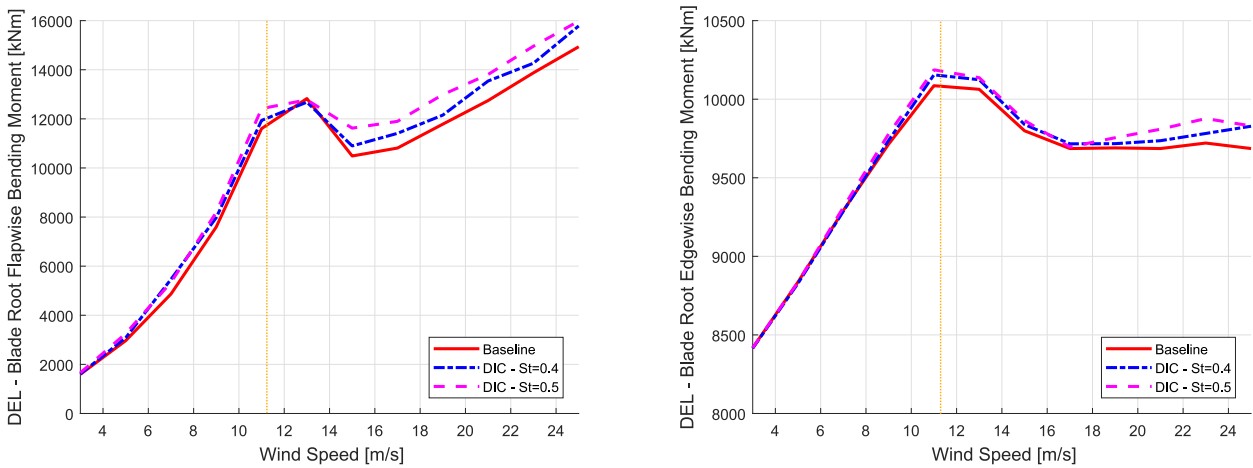

**Figure 8.** Comparison between blade root flap-wise (left) and edge-wise (right) DEL of the baseline (solid red) and DIC with $St = 0.4$ (dash-dotted blue) and $St = 0.5$ (dashed magenta) as functions of mean wind speed. The dashed yellow line indicates the rated wind velocity. Typically, DIC will only be implemented at below-rated inflow velocities.

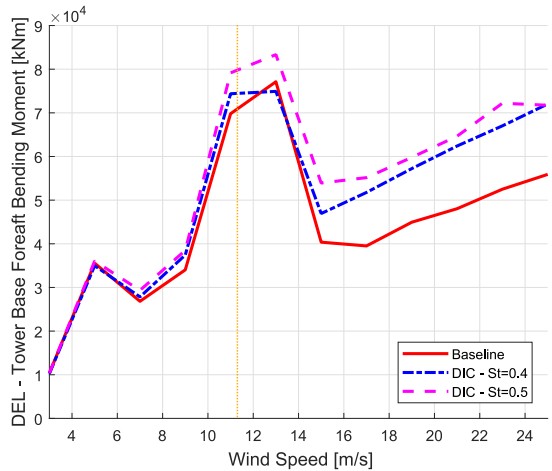 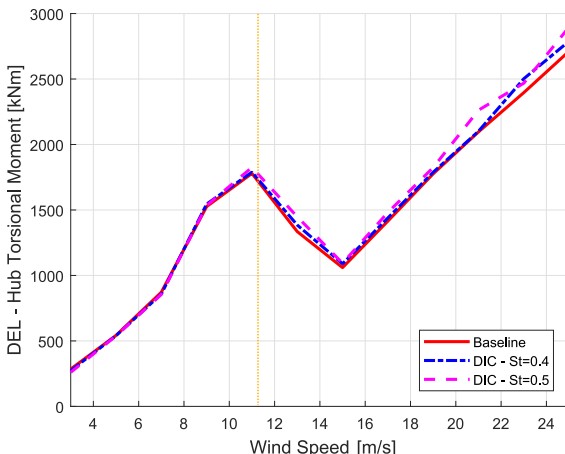

**Figure 9.** Comparison between tower base fore-aft bending moment (left) and hub torsional moment (right) DEL of the baseline (solid red) and DIC with $St = 0.4$ (dash-dotted blue) and $St = 0.5$ (dashed magenta) as functions of mean wind speed. The dashed yellow line indicates the rated wind velocity. Typically, DIC will only be implemented at below-rated inflow velocities.

**Table 3.** Percentage increases of the Weibull-weighted DEL and AEP of the excited turbine compared to the baseline for different Strouhal numbers

|  | Blade Edgewise | Blade Flapwise | Tower ForeAft | Hub Torsion | AEP |
|---|---|---|---|---|---|
| $St = 0.3$ | +0.21% | +2.66% | + 7.06% | +0.94% | -0.46% |
| $St = 0.4$ | +0.40% | +1.80% | + 7.26% | +1.67% | -0.54% |
| $St = 0.5$ | +0.41% | +4.92% | +11.78% | +1.80% | -0.59% |

The Weibull-weighted DEL, $\mathrm{DEL}_w$, is hence computed as

$$\mathrm{DEL}_w = \int_{V_{\mathrm{CI}}}^{V_{\mathrm{CO}}} p_w(V)\,\mathrm{DEL}\,\mathrm{d}V, \tag{2}$$

where $V_{\mathrm{CI}}$ and $V_{\mathrm{C0}}$ are respectively the cut-in and cut-out wind speed.

Considering the class "A", where the Weibull distribution has $k = 2$ and $V_{\mathrm{av}} = 10$ m/s, it is possible to compute the Weibull-weighted DEL for the previously considered loads. To this aim, we suppose to switch off the DIC controller at wind speeds higher than $15\,\mathrm{m/s}$, so that in region III the DELs are lower than the ones shown in the previous figures and equal to the baseline values. These results are summarized in Table 3. As can be seen, the tower base load is affected the most (7 to $11\%$), while loads on the blade flapwise root loads increase with about $2\%$. A negligible impact ($+0.4\%$) is found in the blade edge-wise and in the hub (1 to $2\%$).

So far, the analysis has not considered the probability of activation of the DIC-based wind farm control, which will depend on the specific farm layout and wind rose. From this point of view, the computed DEL increments seen before, as well as the AEP decrease, are to be considered as the worst possible case, as if DIC would always be implemented regardless of wind direction and subsequent wake interaction. It is therefore possible to assess that the impact of DIC on turbine fatigue loads for the analyzed NREL 5 MW reference machine is small compared to the possible gains.

## 6  Experimental Results

In this section, the results of the experiments executed in the wind tunnel at Polimi, as described in Section 4, will be presented. The effects of periodic DIC on the power production of a 3-turbine wind farm are presented for two cases, similar to onshore and offshore wind conditions. The performance of DIC will be compared with the state-of-the-art wind farm control strategies: greedy control, "static" induction control and wake redirection control.

### 6.1  Power production

First, the results with low turbulent wind (TI of approximately 5%) are evaluated. For this case, 3 different sets of experiments have been conducted, as defined in Table 2. These sets each represent one specific amplitude of excitation of the upstream machine: an amplitude of $A = 1$, 1.5 and 2 of $C_T'$ respectively. All other machines operate at their greedy optimum.

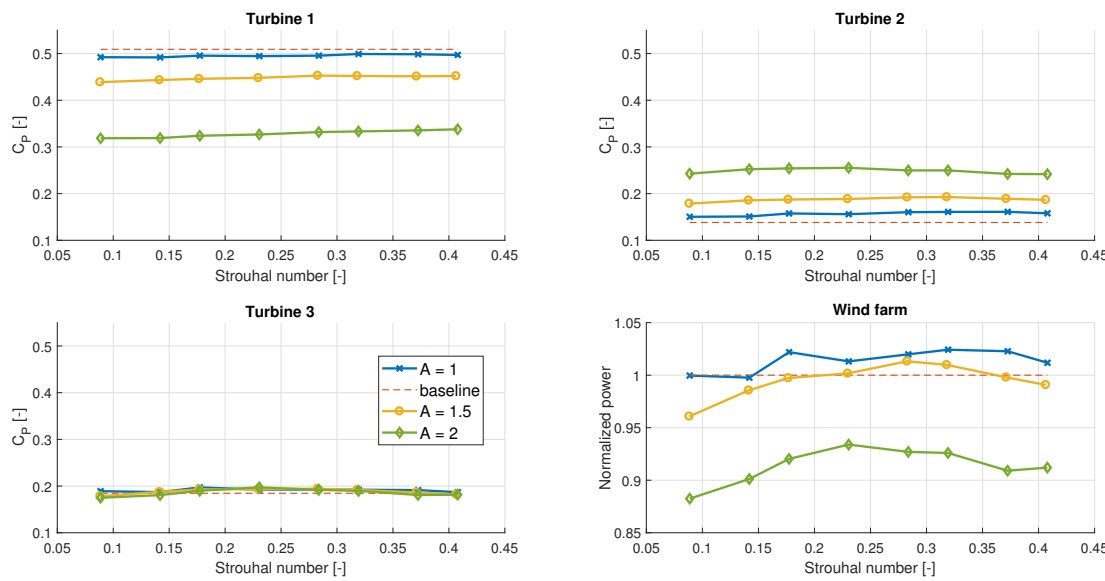

**Figure 10.** $\bar{C}_P$ of the wind farm in low TI conditions for different amplitudes $A$ of $C_T'$, as defined in Table 2. The bottom right figure shows the total power conversion compared to the baseline case.

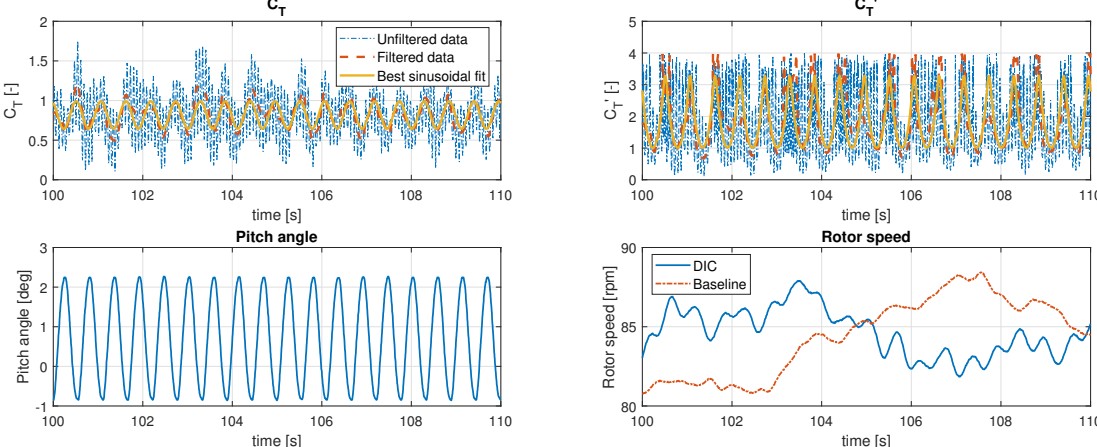

**Figure 11.** Clockwise, the measured $C_T$, $C_T'$, rotor speed and pitch angles of turbine 1 are shown during 10 s of the optimal $St = 0.32$, $A = 1$ DIC experiments in low TI. In the first two figures, the unfiltered data, low-pass filtered data and a best sinusoidal fit are shown. In the fourth figure, the rotor speed during 10 s of the baseline experiment is shown for comparison.

Figure 10 shows the mean power of the turbines and the total wind farm. To account for the small variations in flow conditions, the power is divided by the available power in the wind. As such, these values can be seen as power coefficients. Increasing the amplitude of the sinus decreases the power coefficient of turbine 1, while it increases the power coefficient of the downstream machines. However, for higher $A$, the loss at turbine 1 is too significant to compensate for by the downstream

turbines. The unexpectedly high power loss at turbine 1 could partly be caused by a rotor imbalance that is worsened by higher amplitudes of excitation, leading to significant vibrations of the excited machine. As a result, the case with the lowest amplitude proves to be the most effective.

The highest increase in power extraction is found with $A = 1$ and $St = 0.32$, resulting in a 2.4 % gain. It should be noted that this gain is mostly obtained at turbine 2, while the power at turbine 3 is only marginally higher than in the baseline case.

This corresponds to the conclusions drawn in Munters and Meyers (2018), where a positive effect is observed for turbine 2, but not for machines further downstream. Table 4 gives an overview of the effect of different amplitudes and frequencies on the power production of the 3-turbine model wind farm.

For the sake of reproducibility, Figure 11 shows the measurements of thrust coefficients $C_T$ and $C_T'$, as well as the pitch signal and rotor speed during 10 s of experiments in the optimal control settings ($St = 0.32$, $A = 1$). It should be noted that the

thrust coefficient is obtained by using the definition

$$C_T = \frac{F_T}{0.5\rho A_r U_\infty^2}, \tag{3}$$

where $F_T$ is the thrust exerted on the rotor by the wind, $\rho$ the air density, $A_r$ the rotor area and $U_\infty$ the inflow wind velocity. $F_T$ is determined using the fore-aft bending moment, compensating for tower and nacelle drag, and the pitot measurements in front of turbine 1 (see Figure 5) are used as data for $U_\infty$. This results in a $C_T$-signal disturbed by high frequency noise.

For this purpose, a low-pass filter with a passband frequency of 12.5 Hz was designed. This filter removes the high frequent noise signals, while keeping the excitations caused by DIC (at $f \leq 2.3$ Hz) intact. Furthermore, a sinusoid is fitted on the measurement data using the MATLAB-function LSQCURVEFIT. This function determines the amplitude, offset and phase of the sinusoid that best fit the data. The original data, filtered data and fitted sinusoid are all shown in Figure 11. Finally, the pitch excitation and rotor speed are depicted, the latter clearly showing oscillations caused by DIC. However, these oscillations are relatively small compared to variations caused by changing wind conditions, as the baseline rotor speed shows.

**Table 4.** An overview of the total power increase with respect to the baseline case by applying dynamic induction control with different amplitudes ($A$, rows) and frequencies (columns) for the low TI case.

| Frequency [Hz] | 0.5 | 0.8 | 1 | 1.3 | 1.6 | 1.8 | 2.1 | 2.3 |
|---|---|---|---|---|---|---|---|---|
| **Strouhal** [-] | 0.09 | 0.14 | 0.18 | 0.23 | 0.28 | 0.32 | 0.37 | 0.41 |
| $A = 1.0$ | -0.04% | -0.24% | +2.20% | +1.30% | +1.6% | **+2.4%** | +2.3% | +1.2% |
| $A = 1.5$ | -3.92% | -1.44% | -0.27% | +0.20% | **+1.3%** | +1.0% | -0.20% | -0.92% |
| $A = 2.0$ | -11.76% | -9.89% | -7.97% | **-6.61%** | -7.30% | -7.41% | -9.09% | -8.80% |

Finally, the reliability of these results will be examined. To do this, the results are divided into four segments of 60 seconds. These shorter segments of measurements, still containing 15000 measurement points and between 30 (0.5 Hz) and 138 (2.3 Hz) sine cycles, will then be used to determine the variance of the measurements.

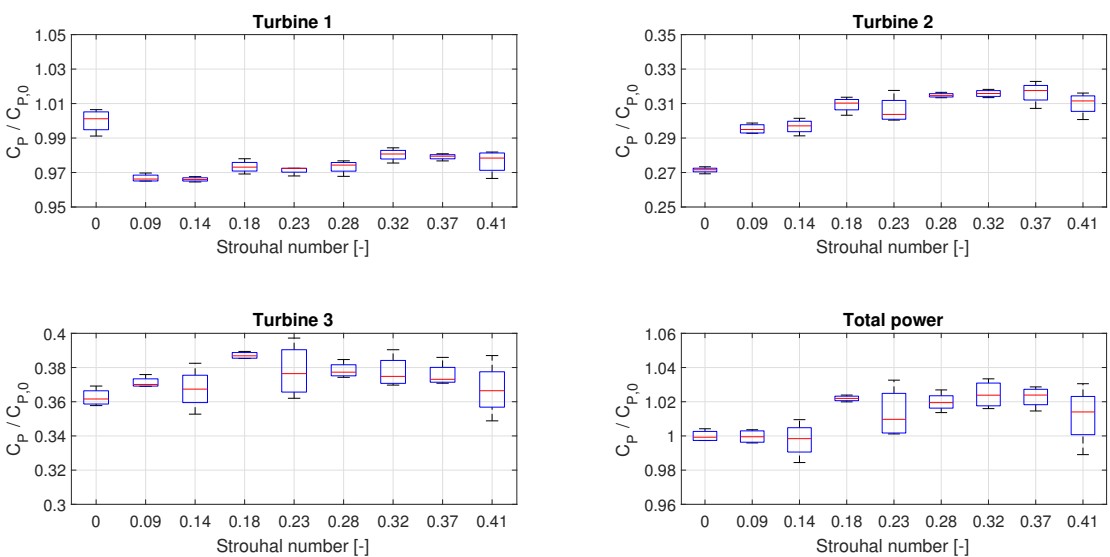

**Figure 12.** A boxplot showing the variance of the $C_P$ measurements for the low turbulent, $C'_T = 1$ experiments, for all turbines individually as well as for the entire wind farm. The $f = 0$ measurement represents the baseline case of no dynamic control.

Figure 12 shows box plots of these data sets for $A = 1$, normalized by the steady state optimal $C_P$ of turbine 1. This figure shows that the variance becomes larger at each downstream row due to the increased turbulence. As a result, the variance is significant in the total power production: up to $\pm 2\%$ of the power. However, this figure also shows that the variance is lower than the power gained by using dynamic induction control: the lowest values of the box plot around the optimal frequency of 5  1.8 Hz are still higher than the baseline value. This analysis therefore indicates that the power increase is significant, as it is not a coincidental result of measurement errors.

**Table 5.** An overview of the total power increase by applying dynamic induction control with different amplitudes ($A$, rows) and frequencies (columns) for the high TI case.

| Frequency [Hz] | 0.5 | 0.8 | 1 | 1.3 | 1.6 | 1.8 | 2.1 | 2.3 |
|---|---|---|---|---|---|---|---|---|
| Strouhal [-] | 0.09 | 0.14 | 0.18 | 0.23 | 0.28 | 0.32 | 0.37 | 0.41 |
| $A = 1.0$ | +1.4% | +1.5% | +2.4% | +1.4% | **+4.0%** | +1.8% | +0.8% | +2.3% |
| $A = 1.5$ | -3.1% | -1.8% | -0.9% | **-0.8%** | -1.0% | -2.3% | -3.4% | -3.6% |
| $A = 2.0$ | -8.9% | -8.7% | -5.2% | -6.7% | -7.7% | **-6.3%** | -8.0% | -8.1% |

Next, the results of the experiments with high turbulence intensity conditions (TI of approximately 10%) will be shown. The results for all the amplitudes and frequencies that were studied are shown in Figure 13. The main conclusion that can be drawn from this figure, is that the effect of exciting the first turbine on the power production of this turbine is lower in these conditions. 10  Due to the turbulence, the baseline power production of this turbine is already slightly lower than in low TI conditions. As a result, the power loss at turbine 1 is negligible for the $A = 1$ case. As the power gain at the downstream turbines is similar, the total power gain for this case is 4%. This gain is found with $A = 1$ and $St = 0.28$, as can be seen in Table 5 where the results are summarized.

When the amplitude of the excitation is increased, the power loss at turbine 1 is comparable with the results in low TI 15  conditions. However, since the power gain at turbine 2 is slightly lower, the total power is also lower than in the baseline case. Subsequently, it seems that the amplitude of the excitation is more important than the frequency in these conditions.

## 6.2 Controller comparison

To emphasize the value of the results shown in the previous subsection, a comparison of the effectiveness of the periodic DIC approach with state-of-the-art wind farm control approaches is executed in the case of full wake interaction. The optimal inputs 20  are found using the steady-state `FLORIS` model (Annoni et al., 2018; Doekemeijer and Storm, 2018), which is calibrated using measurements from the wind tunnel (Schreiber et al., 2017). Three different control strategies are investigated:

– *Greedy control*: all turbines operate at their individual optimum, disregarding wake interaction between turbines.

– *Static induction control*: the induction settings (i.e., collective pitch angles) that predict the highest power capture according to the calibrated `FLORIS` model are implemented.

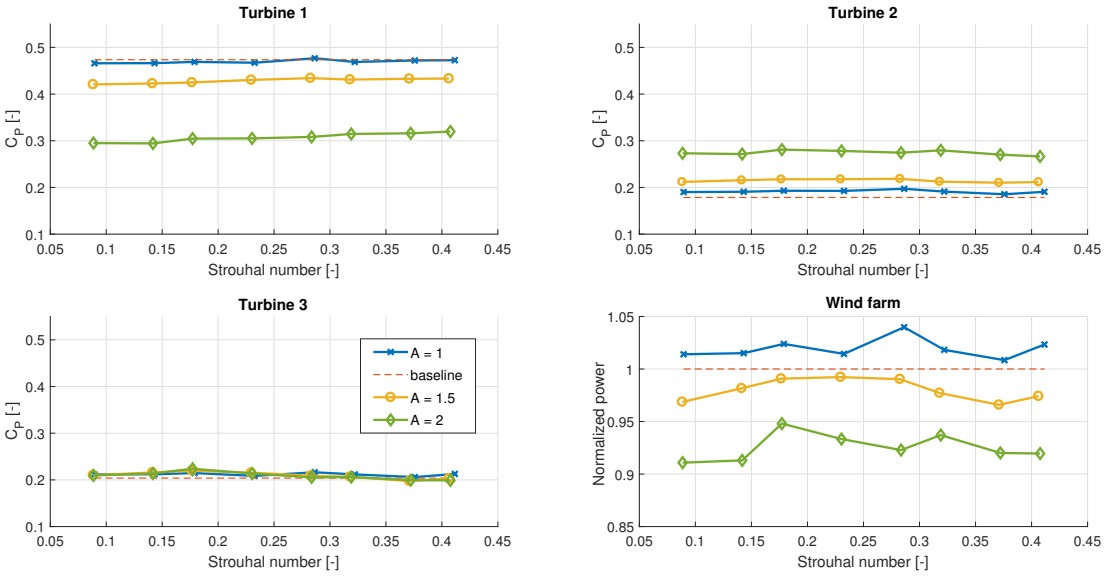

**Figure 13.** $\bar{C}_P$ of the wind farm for different amplitudes $A$ of $C'_T$, as defined in Table 2, in the high TI case. The bottom right figure shows the total power conversion compared to the baseline case.

– *Yaw control*: the yaw angles that predict the highest power capture according to the calibrated `FLORIS` model are implemented.

The results of these experiments are shown in Figure 14. Similar to results in literature (Campagnolo et al., 2016a), static induction control is found to be unable to increase the power production of this wind farm. Yaw control on the other hand
results in a benefit of $3.1\%$ As reported earlier, DIC was able to increase the power production with $2.4\%$ in these conditions. It can therefore be concluded that the potential profit of periodic DIC is significantly higher than with static induction, while it is comparable to that of yaw control when full wake interaction is present.

## 7   Conclusions

In this paper, the effect of periodic Dynamic Induction Control (DIC) on both individual wind turbines and on small wind
farms is investigated. For this purpose, both aero-elastic simulation tools and scaled wind tunnel experiments are used. The unique wind tunnel experiments with DIC show, for the first time, that this control approach not only works in a simulation environment, but also in real world experiments. The results strengthen the results found in simulations executed by Munters and Meyers (2018), showing a potential increase in power production of up to $4\%$, with most of the gain coming from the first downstream turbine. Some minor differences were observed as well. First of all, the optimal Strouhal number is found
to be slightly higher in the wind tunnel experiments, around $St = 0.3$. Secondly, a smaller optimal amplitude of excitation

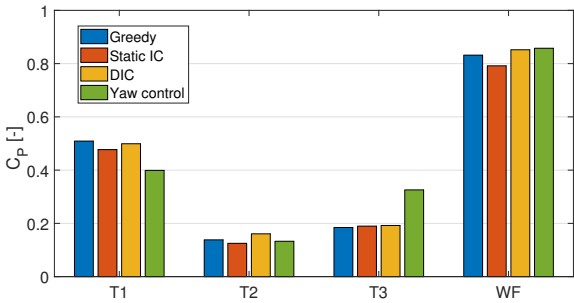

**Figure 14.** The power capture of three state-of-the-art control approaches compared with periodic DIC in low TI conditions. The power capture of the three individual turbines (T1-3), as well the total wind farm (WF) is shown.

was found. This could partly be caused by a slight rotor imbalance, which resulted in significant power losses at the excited turbine. Although higher gains were observed at turbine 2, the power loss of turbine 1 could not be compensated for at higher amplitudes of excitation.

A comparison between DIC and static induction control as well as wake redirection control shows that this approach works
significantly better than the former and approximately as good as the latter. This greatly strengthens the premise that DIC is an effective method to increase the power production of a wind farm as a whole. Furthermore, by means of the aeroelastic tool CP-LAMBDA, it was shown that the effect of DIC on the Damage Equivalent Loads (DEL) of the excited wind turbine is relatively small. For the given wind turbine example, the weighted blade root edgewise DEL was in the order of $0.3$ to $0.4\%$ higher than in the baseline greedy control case.

In all, it can be concluded that the dynamic induction control approach shows great promise, as now both simulations and scaled experiments show that it is possible to achieve a power gain. However, some minor differences are found between simulation studies in literature and the experiments presented here, which still need to be adressed. Future research can therefore be directed into clarifying these differences, as well as executing additional experiments, for example with different inflow velocities inside and outside the region II regime.

As the amplitude and frequency of the excitation are shown to be important control parameters, it would be a very interesting challenge to develop an algorithm that is able to optimize these parameters. Furthermore, additional analysis on the increased loads on the (downstream) turbines can be done to investigate the effect of these loads on the lifetime of turbines, as well as the tradeoff between power and load effects. Another possible approach would be to investigate the effects of applying periodic DIC on intermediate wind turbines on the performance of the wind farm. Finally, application on full-scale wind turbines could
be the last step in proving the validity of this approach.

*Acknowledgements.* This work has been supported by the CL-Windcon project, which receives funding from the European Union Horizon 2020 research and innovation programme under grand agreement No. 727477.

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
