# Peer review of "Periodic dynamic induction control of wind farms: proving the potential in simulations and wind tunnel experiments"

_Wind Energy Science, 2019_

## Referee Comment (RC1) · Anonymous Referee #1 · 26 Aug 2019

The paper is well presented and well argued, and adds valuable contributions to the literature, including a first analysis of loads in dynamic induction control as well as a wind tunnel study validating the approach. The figures and descriptions are good, and the paper is very direct to understand.

Mostly minor comments follow below. Main over-arching comment is really a question to propose be considered in the next version of the paper. Fig 9 shows very small effect on turbine 3. Is this to be expected? In completing this review, I re-read "Towards practical dynamic induction control of wind farms: analysis of optimally controlled wind-farm boundary layers and sinusoidal induction control of first-row turbines" and found

this passage:

—— Figure 8 illustrates that the first-row optimized thrust coefficient also results in a significant power increase in the third row, which is not observed using the sinusoidal thrust strategy. Furthermore, the analysis of the modified control cases in Fig. 11 proves that the first-row controls are also partially synchronized with the flow. This shows that other mechanisms, dependent on specific flow events for increasing wind-farm power, are at play as well. Even though the application of regression algorithms in an attempt to link turbine actions to low-dimensional flow measurements (e.g., local velocity, shear and kinetic energy) has been unsuccessful thus far, similar analysis based upon more complex flow features (e.g., vorticity structures, high-speed turbulent streaks, or downdrafts) might be more promising. This requires further optimal control simulations over an extended time, as the total control time horizon of 30 min in the current dataset is insufficient for robust statistics in this kind of analysis. This is an important remaining challenge to be addressed in future research. ——

As well as this from the conclusion of the same paper:

—— Although the first-row sinusoidal control led to a robust increase in total power for a reduced-size 4×4 wind farm, a full-scale test indicated that downstream turbine activity is required to obtain increased power at larger farm scales. It was also shown that the simple sinusoidal strategy does not lead to increased power extraction when applied to downstream intermediate turbines. Identifying the mechanisms for power increase in these turbines hence remains an important open research question. ——

My reading is that yes, these results do confirm this, the third turbine is not expected to increase in power unless (if I understand correctly) 1. The first turbine pursues a non-sinusoidal DIC or 2. The second turbine performs DIC additionally

Do you agree? Are there plans to try any DIC on the second turbine etc?

Small Comments:

Fig 1 could use a more descriptive labeling/caption, it's not clear what each of the lines represent

DTU 5 MW turbine (Jonkman et al., 2009) — shouldn't that be NREL? 5MW (based on reference provided)

Table 1, for experiments the control input is Beta, but amplitude is specified in Ct? (Now I see this is explained later in the text, but might be good to ensure the explanation is indicated in the table or indicate to the reader explanation is coming?)

Figure 6: This is a really useful view into the loading impacts

Is there a reference for Weibull-weighted DELs? A nice idea, are they used often?

Fig 7-8, why do the effects persist above 15 m/s? I believe this addressed in text, but could be useful to re-iterate in caption, maybe also indicate with a vertical line where the DIC would be actually shut off?

Fig 8: seems to have an error in caption

Section 6.2 Do you use the FLORIS model of Gebraad 2016, or the newer gaussian model of Bastankah within FLORIS? Maybe provide FLORIS version number?

---

## Referee Comment (RC2) · Johan Meyers (Referee) · 5 Sep 2019

Very interesting work, which I strongly recommend for publication. I have a number of smaller comments, that should be relatively easy to incorporate in a revision.

1. abstract: "In this paper, only periodic variation, " –> variations

2. Figure 1: please improve. In 1a (bottom) – for clarity, please indicate levels of $C\_T$ associated with different velocity profiles. In 1b, not clear what the order is of the velocity profiles (in time or phase of the sinusoidal forcing). Also not 100% convinced that this will be the effective response – is this an 'artists' impression, or is this based

on some model? Please clarify in the fig caption and text.

3. In the paper, it is suggested a couple of times that CFD is performed:

- Page 2: "Simulations will be executed using the high-fidelity Computational Fluid Dynamics (CFD) environment SOWFA"

- Page 7: "Once the optimal DIC parameters in terms of wake mixing have been evaluated using CFD, . . ." However, apart from these, CFD seems not to be really discussed. . . Please clarify. If you use CFD in some way, it would merit a much lengthier description (computational domain, mesh, boundary conditions, models used, some results, . . .)

4. Figure 2: how was this figure constructed (please make caption more self-contained). Did you use the procedure described on top of page 4? Or did you use BEM, or the Cp-Lambda model, . . .

5. page 4: "A region I-1/2 with constant rotor speed equal to 6 rpm extends from the cut-in speed of 4 m/s to 7 m/s." I'm a bit surprised by this – please double check. As far as I remember, in region 1.5 the rotor speed is increasing, and not constant .

6. Table 2: for completeness, please add values for average pitch angle and amplitude of pitch oscillation

7. Following up on previous point, for sake of reproducibility, it would make sense to add a detailed figure with the C_T & C_T' signal together with the pitch signal and the rotational speed signal

8. page 7,line 15: "Once the optimal DIC parameters in terms of wake mixing have been evaluated using CFD, . . ." not sure CFD is used. . . - cf point 3 above? How did you determine optimal DIC parameters?

9. page 8, line 9: please refer again to Turbsim, and IEC when you reference to NTM

10. Figure 8, check caption

11. page 11, start of section 6.1: five different cases are mentioned, but later on, results of only three experiments seem to be reported (the ones with different amplitudes). What about results for block signal, and results for phase difference between turbines?

12. Figure 9: I'm a bit confused: in the caption you mention different amplitudes, but in the legend (bottom-left panel) you seem to show averaged values for $C_T$ (1, 1.5, 2). First of all – are these averaged values of $C_T'$ (see table 2)? Therefore, do you mean different average & amplitude. Please clarify and improve caption/legend

13. page 15, line 4: "It can therefore be concluded ...". In the work of Munters, Sinusoidal DIC was shown to work for the first turbine, with a positive effect on the second, but not on the third. Sinusoidal DIC applied to the second (or later) turbines did not work. The results in the current paper seem to confirm this. Therefore, this conclusion should probably be adapted/tuned down a bit + maybe additional discussion on future work in the conclusions section.

14. Continuing on the previous point: what about the results of the out-of-phase experiment with the first & second turbine (cf. comment 11 above) – was this intended to improve turbine 3 performance – if so, what were the results. Did you do in-phase as well? Reading the text, I'm presuming that most experiments were only using sinusoidal DIC on the first turbine? Is that correct? Should maybe be emphasized/discussed a bit more throughout.

15. page 15, line 15: to be fair, you should compare weighted DEL against weighted power gain (which will also be much lower when averaged over a Weibull distribution)

16. page 16, line 1: significant differences between simulations and experiments. What do you mean by that? please clarify. . .

---

## Referee Comment (RC3) · Anonymous Referee #3 · 10 Sep 2019

The paper is well structured and makes a relevant contribution with first scaled wind tunnel experiments of dynamic induction farm control, as well as load evaluation by aeroelastic simulation for excited upstream wind turbine. Sound methodology is applied to results analysis. Publication is recommended upon addressing some minor comments listed below, added to those of the other referees.

* Page 8, Line 1 -> Which was the reason behind the choice of a pitch amplitude of 2 degrees? Could you please better specify? Has this pitch amplitude any relation to the amplitude used in the scaled tests? Besides, the experiments have shown greater dependency on the amplitude than on the frequency (Strouhal number). Wouldn't it

be coherent to perform in future work the load simulations also in accordance to this by varying the pitch amplitude in order to see the effect on loading of changing such amplitude?

* Section 7- Conclusions could be further elaborated by gathering nice comments previously included in the paper and by precising better some aspects:

-> It is shown that by acting on turbine 1, turbine 3 remains unaffected.

-> It is shown that, for a given mean wind speed, the change in the power gain mostly depends on the amplitude of the DIC and not on the frequency. Would it be any dependence on the mean wind speed? The experiments have examined the effect of DIC under different TI conditions. It would also be interesting to see in the future the effect under different mean wind speed conditions.

-> Page 15, Line 17 to Page 16, Line 1: "In all, it can be concluded that the dynamic induction control approach shows great promise, as now both simulations and scaled experiments show that it is possible to achieve a power gain. However, significant differences are found between simulation and experiments, which still need to be addressed." -> The conclusion included does not apply to the presented simulation results, which consist in the simulation of one single turbine, mainly for loading evaluation. These simulations don't provide insights into the behavior and power gain at farm level. Equally, it is not clear which are the significant differences between simulation and experiments this statement makes reference to.

* Is there any hypothesis on why the increase in the DIC amplitude provokes such decrease in the final power gain?

* For practical application of the technology, taking into account that DIC is intended for region II -among others-, have you considered the possible risk of stall when applying a periodic pitch variation of several degrees around fine pitch? The value of 2 degrees used in simulations (section 5) could prove to be relevant.

* The lowest tested amplitude for DIC has proved to be the best one. So, one question that arises is whether further decrease in the amplitude would lead to even better results. It would be interesting to determine in the future which is the minimum "A" that provides the maximum power gain.

* In the wind tunnel experiments it has been possible to measure the thrust coefficient thanks to the knowledge about the wind conditions. This has allowed the determination by trial and error of the pitch variation in order to provide a thrust coefficient (amplitude, frequency) matching the desired one. How would this technology be applicable in real wind turbines where such detail of information about wind conditions is not so easily and precisely available?

————————————————————————————————————————————————- For the sake of clarity and reproducibility:

* It would be advisable to indicate upfront from the very beginning of the paper that it focuses on below rated conditions and excitation of collective pitch angle. Also, to leave an explanatory comment about induction as in-wake speed deficit.

* Table 1: Missing frequency units in last row ("Frequency of excitation in St"). It's understood that it is "Hz", but better to leave it explicit.

* Table 2: Please make coherent the denomination for the amplitude variable A (third column in the table) with the description in the table caption (CT,DIC).

* Page 7, Line 18 -> It could be added as examined load the "hub torsional moment", taking into account that these results are presented in Table 3.

* Page 8, Line 9 -> It could be added "mean" therefore indicating "mean hub wind speed of"

* Figure 7 and Figure 8, caption -> It could be added "mean" therefore indicating "mean wind speed"

* Table 3. The table caption would be clearer if it is indicated that the percentages refer to improvement with respect to baseline. Equally, it is indicated "AEP" in the caption, although the values are not included in the table. The percentage of variation of power with respect to baseline is of great interest, in order to compare the order of magnitude with the results of turbine 1 in the wind tunnel experiments. So, it would be advisable to introduce such information, not only in terms of AEP, but also through a figure of comparison with baseline, for example power time plot corresponding to Figure 5.

* Section 6. It would be advisable to indicate the layout of the wind farm tested in the wind tunnel, either through written explanation or through a descriptive figure.

* Table 4, caption -> Caption could be clearer by making reference to baseline: "An overview of the total power increase with respect to baseline by applying"

* Table 4 and Table 5 -> It would be advisable to indicate the frequency units (first row).

* Page 11, Line 5 -> When mentioning the change of +2% in blade root loads, it would be advisable to specify "flapwise". Equally, when mentioning the negligible impact found in edge-wise and in the hub, it would be clearer to mention the respective percentages, since for edgewise, it's only 0.4%, but for the hub it accounts for 1% to 2%. The discussion of load results is mainly done for St = 0.4 and St = 0.5, while the best fit for experiments is provided by St = 0.33 (low TI) and St = 0.29. Which would be the correspondence between the St results in the scaled tests and those for a full-scale model such as the one simulated in CP-LAMBDA?

* Page 11, Line 18 -> When making reference to the experiments with different amplitudes on a sinusoidal input, it would be convenient to introduce the reference to Table 2. Equally, it could be helpful to indicate again that the sinusoidal input is "applied to the collective pitch", which is the range of variation of the pitch angle, and which correspondence this would have with the pitch angle in a full-scale wind turbine.

* Page 13, Line 3. In the same way that it is indicated explicitly for low TI experiments

(Page 11, Line 17), it would be nice to indicate the approximate value of TI applied in the high TI experiments.

* Page 13, Line 6. For higher clarity, it could be indicated to which production it makes reference the sentence. It is understood that it refers to: "the baseline power production of this turbine is already slightly lower than in low TI conditions".

* Page 14, Line 8 -> For the sake of clarity, it would be advisable to introduce again the reference "Schreiber et al. (2017)", which was already indicated in Page 4.
* * *
* Page 3, line 8 -> "were" instead of "where"

* Table 1 The frequencies of excitation in St indicated for the aeroelastic simulations "Between 0.3 and 0.5" don't match the range of frequencies of DIC stated in Section 5, Page 8, where it is stated that this frequency varies from 0.00952 Hz to 0.0595 Hz. Equally, the frequencies indicated for the experiments [0.09-0.41] don't match the frequencies included in Table 4 and Table 5 [0.5-2.3].

* Page 6, line 15 -> "kHz" instead of "kH"

* Figure 5, xlabel -> It would be preferable to indicate time units in accordance to the symbol stated by the International System of Units: "s"

* Figure 7 and Figure 8, xlabel -> It could be introduced a space between Wind Speed and the unit [m/s]

* Page 11, Line 1 -> According to SI unit rules and style conventions, unit should not be italic "m/s".

* Page 11, Line 3 -> In accordance to style convention, there should be a space between the number and unit "15 m/s"

* Page 11, Line 22 -> It seems that the verb is missing in the sentence: "the power is

divided"

* Figure 9, Caption -> The reference in the figure legend and caption should be coherent between CT and C'T.

* Figure 11, legend -> It seems that "baseline" would fit better than "benchmark", also keeping coherence with previous figures such as Figure 9.

* Page 14, Line 2 -> It seems that the sentence "However, since the power gain at turbine 3 is slightly lower, the total power is also lower than in the baseline case" would indeed make reference to turbine 2, according to the figures.

* Page 15, Line 15 -> To be corrected "weighted" instead of weighed. It would be preferable to specify "the increase of the weighted DEL with respect to baseline". Equally, the values of DEL included could be misleading without specifying which load they make reference to. Indeed, the 0.3-0.4% refers to blade root edgewise, which is the least affected by DIC.

---

## Referee Comment (RC4) · Anonymous Referee #4 · 10 Sep 2019

Dear authors,

Thank you very much for submitting the paper to the WES journal. It was nice reading the paper and it is of high quality. Altogether a lot of relevant work is presented and it gives a significant contribution to the community. The paper follows a clear structure and gives a lot of background information that helps to understand the tasks that have been performed. Altogether I recommend the publication with the consideration of the following minor corrections and the comments of the other reviews.

Abstract: Please introduce the idea of induction control before naming it and extend the abstract a litte more. This would help people being not familiar with the topic to

unterstand the content of the paper. Figure 1: Please explain the figure in more detail in the caption. This figure basically presents the whole concept and needs therefore more explanation. p. 2 l.4: you say DTU 5 MW turbine -> NREL 5 MW turbine Table 1: Munters et. al. Table 1: please first introduce beta and cT' before having the table. I know that latex is placing it like this, but moving it to the next page is preferable. Page 4: yaw control: to me wake steering is more familiar than yaw control. Maybe you need to add both or replace it Figure 7-12: the style of the labels differ to the previous plots, Figure 7, 8: a space before unit (As mentioned in caption Fig. 8) Conclusions: p.16 l.1: please again name the differences in the conclusions Acknowledgements: program -> programme

---

## Author Comment (AC1) · 18 Oct 2019

| | |
|---|---|
| Date | October 18, 2019 |
| Our reference | WES-2019-50 |
| Contact person | J.A. Frederik |
| Telephone/fax | +31 (0)15 27 85623 / n/a |
| E-mail | J.A.Frederik@TUDelft.nl |
| Subject | Author's response |

**Delft University of Technology**

Delft Center for Systems and Control

Address
Mekelweg 2 (3ME building)
2628 CD Delft
The Netherlands

www.dcsc.tudelft.nl

Referees
*Wind Energy Science Discussion*

Dear Reviewers,

First of all, the authors would like to thank the reviewers for their positive and constructive feedback. We believe that the comments have helped us to further improve the quality of the paper. In our attempt to account for the comments, we have revised different aspects of the paper. The objective of this document is to respond to the points raised by the reviewers and to provide a detailed overview of the changes made to the paper. In the subsequent sections, we will respond to the review report provided by each of the reviewers.

Yours sincerely,

Joeri Frederik

Enclosure(s): Response to comments of Anonymous Referee #1
Response to comments of Johan Meyers
Response to comments of Anonymous Referee #3
Response to comments of Anonymous Referee #4

**Response to comments of Anonymous Referee #1**

- The paper is well presented and well argued, and adds valuable contributions to the literature, including a first analysis of loads in dynamic induction control as well as a wind tunnel study validating the approach. The figures and descriptions are good, and the paper is very direct to understand.

  The authors would like to thank the referee for the positive feedback.

- Mostly minor comments follow below. Main over-arching comment is really a question to propose be considered in the next version of the paper. Fig 9 shows very small effect on turbine 3. Is this to be expected? In completing this review, I re-read "Towards practical dynamic induction control of wind farms: analysis of optimally controlled windfarm boundary layers and sinusoidal induction control of first-row turbines and found this passage: Figure 8 illustrates that the first-row optimized thrust coefficient also results in a significant power increase in the third row, which is not observed using the sinusoidal thrust strategy. Furthermore, the analysis of the modified control cases in Fig. 11 proves that the first-row controls are also partially synchronized with the flow. This shows that other mechanisms, dependent on specific flow events for increasing windfarm power, are at play as well. Even though the application of regression algorithms in an attempt to link turbine actions to low-dimensional flow measurements (e.g., local velocity, shear and kinetic energy) has been unsuccessful thus far, similar analysis based upon more complex flow features (e.g., vorticity structures, high-speed turbulent streaks, or downdrafts) might be more promising. This requires further optimal control simulations over an extended time, as the total control time horizon of 30 min in the current dataset is insufficient for robust statistics in this kind of analysis. This is an important remaining challenge to be addressed in future research. As well as this from the conclusion of the same paper: Although the first-row sinusoidal control led to a robust increase in total power for a reduced-size 44 wind farm, a full-scale test indicated that downstream turbine activity is required to obtain increased power at larger farm scales. It was also shown that the simple sinusoidal strategy does not lead to increased power extraction when applied to downstream intermediate turbines. Identifying the mechanisms for power increase in these turbines hence remains an important open research question. My reading is that yes, these results do confirm this, the third turbine is not expected to increase in power unless (if I understand correctly) 1. The first turbine pursues a non-sinusoidal DIC or 2. The second turbine performs DIC additionally Do you agree? Are there plans to try any DIC on the second turbine etc?

Figures 9-11 in the paper show that, in the wind tunnel, turbine 3 does in fact have a slightly increased power production when periodic DIC is applied on turbine 1. This gain is very small - much smaller than the gain obtained at turbine 2 - but as Figure 10 shows, it is in fact significant. Therefore, the claim that *no* power increase at turbine 3 is expected with periodic DIC is therefore not supported by the data presented in this paper. However, to address this point more specifically in the paper, both in the analysis in Section 6.1 and in the Conclusions, it will be stressed that the majority of the gain in power production is obtained at turbine 2. With regards to periodic DIC on the second turbine: we have in fact executed wind tunnel experiments with periodic DIC on both the first and the second turbine. However, the results of these experiments are as of yet inconclusive, which is why they are not included in this paper. Future research in this topic would definitely be of interest to us, although there are no direct plans for this. For completeness, this research direction is added to the future research opportunities in Section 7.

- Small Comments: Fig 1 could use a more descriptive labeling/caption, its not clear what each of the lines represent

  A more descriptive caption is added to Figure 1 to explain more elaborately what is shown in this figure: *A schematic representation of a wind turbine in flow field, showing the working principles of static (a) and dynamic induction control (b). On the top, the turbine is simplified as a rotor disk, and its streamtube - the area where the wind speed is affected by the turbine settings - is depicted. The force $F_T$ exerted on the wind is shown for different induction settings, where red depicts greedy control, orange and yellow arbitrary static derating settings, and green periodic DIC. The bottom figures show the corresponding wind velocity profiles, with respect to inflow velocity $U_\infty$, as a function of the distance from the turbine. The area highlighted in blue is where a downstream turbine is typically located.*

- DTU 5 MW turbine (Jonkman et al., 2009), shouldnt that be NREL? 5MW (based on reference provided)

  The referee is absolutely right. This erratum has been corrected as suggested.

- Table 1, for experiments the control input is Beta, but amplitude is specified in Ct? (Now I see this is explained later in the text, but might be good to ensure the explanation is indicated in the table or indicate to the reader explanation is coming?)

  To clarify the effect of $\beta$ on $C'_T$, the following sentence is added in the caption of Table 1: *Note that the pitch amplitude $\beta = 2°$ used in the simulations leads to a amplitude of approximately $C'_T = 1.5$.*

- Figure 6: This is a really useful view into the loading impacts Is there a reference for Weibull-weighted DELs? A nice idea, are they used often?

XXX

- Fig 7-8, why do the effects persist above 15 m/s? I believe this addressed in text, but could be useful to re-iterate in caption, maybe also indicate with a vertical line where the DIC would be actually shut off?

  As mentioned in the text, "The DIC was assumed to be activated for wind speeds between 3 and 25 m/s, to cover the totality of regions I-1/2, II, II-1/2 and III", which "is to be regarded as a conservative choice". When DIC is only applied in region II, the loads will of course be identical to the baseline case above rated wind speeds. To further emphasize this, a vertical line is included to indicate the rated wind speed, with the caption describing that "Typically, DIC will only be implemented at below-rated inflow velocities."

- Fig 8: seems to have an error in caption

  There is indeed an error in the caption, which has been removed.

- Section 6.2 Do you use the FLORIS model of Gebraad 2016, or the newer gaussian model of Bastankah within FLORIS? Maybe provide FLORIS version number?

  The FLORIS model with gaussian distribution as proposed by Bastankah was used. For clarity, the reference was changed to represent the version that was used in this paper.

**Response to comments of Johan Meyers**

- Very interesting work, which I strongly recommend for publication. I have a number of smaller comments, that should be relatively easy to incorporate in a revision.

  The authors would like to thank prof. Meyers for the kind words, and hope to address all the smaller comments to his satisfaction.

- 1. abstract: In this paper, only periodic variation, ¿ variations

  This erratum has been corrected as suggested.

- 2. Figure 1: please improve. In 1a (bottom) for clarity, please indicate levels of $C_T$ associated with different velocity profiles. In 1b, not clear what the order is of the velocity profiles (in time or phase of the sinusoidal forcing). Also not 100% convinced that this will be the effective response is this an artists impression, or is this based on some model? Please clarify in the fig caption and text.

  To answer the final question posed by the referee: this figure is not based on some model or measurement, but rather a schematic representation of the flow through a rotor streamtube, meant only to clarify the working principle of DIC with respect to static induction control. As such, the lines do not represent specific values of $C_T$ or $U_\infty$. To emphasize this, a more elaborate caption has been added to this figure: *A schematic representation of a wind turbine in flow field, showing the working principles of static (a) and dynamic induction control (b). On the top, the turbine is simplified as a rotor disk, and its streamtube - the area where the wind speed is affected by the turbine settings - is depicted. The force $F_T$ exerted on the wind is shown for different induction settings, where red depicts greedy control, orange and yellow arbitrary static derating settings, and green periodic DIC. The bottom figures show the corresponding wind velocity profiles, with respect to inflow velocity $U_\infty$, as a function of the distance from the turbine. The area highlighted in blue is where a downstream turbine is typically located.*

- 3. In the paper, it is suggested a couple of times that CFD is performed: - Page 2: Simulations will be executed using the high-fidelity Computational Fluid Dynamics (CFD) environment SOWFA - Page 7: Once the optimal DIC parameters in terms of wake mixing have been evaluated using CFD, ... However, apart from these, CFD seems not to be really discussed... Please clarify. If you use CFD in some way, it would merit a much lengthier description (computational domain, mesh, boundary conditions, models used, some results, ...)

The CFD simulations mentioned here were removed from the paper in one of the final stages before submission. The most important reason for this was that the authors felt like the contribution of this CFD study to the already existing literature (mostly by Munters and Meyers) was limited. We therefore chose to focus on the most important scientific contributions: the load analysis and the wind tunnel experiments. All references to CFD simulations have been removed in the updated version of the manuscript.

- 4. Figure 2: how was this figure constructed (please make caption more selfcontained). Did you use the procedure described on top of page 4? Or did you use BEM, or the Cp-Lambda model, ...

  This figure was constructed using look-up tables based on data from the $G1$ turbine models. For clarity, this has been added to the caption: *Values of $C_T$ for different types of input signals, created using a look-up table for the G1 turbine model. The thrust coefficient is shown for three different sinusoidal excitations: on $C_T$, on $C'_T$ and on the collective pitch angle $\beta$, tuned such that the amplitude of $C'_T$ is 1.5. The dashed line shows the steady-state optimal $C_T$.*

- 5. page 4: A region I-1/2 with constant rotor speed equal to 6 rpm extends from the cut-in speed of 4 m/s to 7 m/s. Im a bit surprised by this please double check. As far as I remember, in region 1.5 the rotor speed is increasing, and not constant.

  XXX

- 6. Table 2: for completeness, please add values for average pitch angle and amplitude of pitch oscillation

  As suggested by the referee, mean values of the average and amplitude of the pitch angle are added to Table 2.

- 7. Following up on previous point, for sake of reproducibility, it would make sense to add a detailed figure with the $C_T$ & $C'_T$ signal together with the pitch signal and the rotational speed signal

  As requested by the referee, such a figure has been added to Section 6. The figure shows the requested variables for the optimal low-TI case: $St = 0.31$, $A = 1$. The $C_T$ and $C'_T$ measurements are displayed, both filtered and unfiltered, as well as the best sinusoidal fit to this data. Furthermore, the pitch excitation and the rotor speed is given, with the latter also compared to the baseline case.

- 8. page 7,line 15: Once the optimal DIC parameters in terms of wake mixing have been evaluated using CFD, ... not sure CFD is used... - cf point 3 above? How did you determine optimal DIC parameters?

As explained in point 3, the CFD simulations were removed from the paper. The parameters chosen here are close to the optimum found in the wind tunnel.

- 9.  page 8, line 9: please refer again to Turbsim, and IEC when you reference to NTM

The references suggested by the referee have been added here.

- 10.  Figure 8, check caption

The erratum in the caption has been removed.

- 11.  page 11, start of section 6.1: five different cases are mentioned, but later on, results of only three experiments seem to be reported (the ones with different amplitudes). What about results for block signal, and results for phase difference between turbines?

The results of these last two experiments have been cut from the paper, since the results were as of yet inconclusive. However, the authors have overlooked this reference to these experiments, which was therefore not removed. This has been done now.

- 12.  Figure 9: Im a bit confused: in the caption you mention different amplitudes, but in the legend (bottom-left panel) you seem to show averaged values for $C_T$ (1, 1.5, 2). First of all  are these averaged values of $C_T$ (see table 2)? Therefore, do you mean different average & amplitude. Please clarify and improve caption/legend

This figure shows, as mentioned in the caption, results for different amplitudes of excitation of $C_T'$. To remove any ambiguity, the legend has been changed to read Amplitude $A$ instead of $C_T$. Furthermore, a reference to Table 2 is added, where the corresponding mean and amplitude of $C_T$ and pitch angle $\beta$ can be found.

- 13.  page 15, line 4: It can therefore be concluded .... In the work of Munters, Sinusoidal DIC was shown to work for the first turbine, with a positive effect on the second, but not on the third. Sinusoidal DIC applied to the second (or later) turbines did not work. The results in the current paper seem to confirm this. Therefore, this conclusion should probably be adapted/tuned down a bit + maybe additional discussion on future work in the conclusions section.

This comment is very similar to the first comment of Referee #1. For a more detailed response, the reader is therefore referred to the response given here. In short, the wind tunnel experiments show that the largest positive effect is measured at turbine 2, but there is also a (very small) positive effect at turbine 3. A more elaborate discussion on these results has been added to both Section 6.2 (results) and 7 (conclusions).

- 14. Continuing on the previous point: what about the results of the out-of-phase experiment with the first & second turbine (cf. comment 11 above)  was this intended to improve turbine 3 performance  if so, what were the results. Did you do in-phase as well? Reading the text, Im presuming that most experiments were only using sinusoidal DIC on the first turbine? Is that correct? Should maybe be emphasized/discussed a bit more throughout.

  First of all: yes, it is correct that in the results presented in this paper, periodic DIC was only applied on the first (upstream) turbine. To emphasize this, a mention of this is added once more both in Section 2 (Control Strategy) and Section 6 (Results).
  Secondly, regarding the experiments with periodic DIC on both turbines 1 and 2: as mentioned at the response to comment 11, these results were inconclusive. Based on the experiments, it could not be said whether this strategy would positively effect the power capture of the wind farm, nor what the influence of a phase offset was. Therefore, the choice was made not to include these results in this paper. This is possible future research direction though, and as such has been added to the conclusions.

- 15. page 15, line 15: to be fair, you should compare weighted DEL against weighted power gain (which will also be much lower when averaged over a Weibull distribution)

  The referee is absolutely right that the power gain weighted over a Weibull distribution would be significantly lower, as periodic DIC will only be effective when there is full wake interaction between turbines. However, this paper does not investigate the potential AEP of a wind farm. Rather, it shows that - when wake interaction is present - periodic DIC can be an effective method to increase power production, with the load effects being relatively small. As already mentioned in the conclusions, a future research challenge lies in further investigating the turbine loads with respect to the potential power gain.

- 16. page 16, line 1: significant differences between simulations and experiments. What do you mean by that? please clarify...

  There are some differences between the results found in simulations executed by Munters and Meyers, and the wind tunnel results presented in this paper. Most notably, the optimal frequency and amplitude of excitation is found to be slightly higher and lower respectively. To name these differences "significant" might be a bit too definite, so this was changed to "some minor differences". Furthermore, the aforementioned differences are now explicitly named in a prior paragraph of the conclusions.

**Response to comments of Anonymous Referee #3**

- The paper is well structured and makes a relevant contribution with first scaled wind tunnel experiments of dynamic induction farm control, as well as load evaluation by aeroelastic simulation for excited upstream wind turbine. Sound methodology is applied to results analysis. Publication is recommended upon addressing some minor comments listed below, added to those of the other referees.

  The authors would like to thank the referee for his constructive feedback in improving the quality of the paper.

- Page 8, Line 1: Which was the reason behind the choice of a pitch amplitude of 2 degrees? Could you please better specify? Has this pitch amplitude any relation to the amplitude used in the scaled tests?

  The pitch amplitude of 2 degrees leads, for the NREL 5MW turbine, to an excitation amplitude of $C_T'$ of approximately $A = 1.5$. This case can therefore be considered an "average" load case. This clarification is now added to Table 1, where the different cases are defined.

- Besides, the experiments have shown greater dependency on the amplitude than on the frequency (Strouhal number). Wouldnt it be coherent to perform in future work the load simulations also in accordance to this by varying the pitch amplitude in order to see the effect on loading of changing such amplitude?

  The authors agree that this would be a very interesting future research direction. The analysis presented in this paper should really be seen as a first step in evaluating the load effects of DIC. Such an investigation would indeed be very interesting to perform. Further investigation into these loads has been added more explicitly to the future research possibilities in Section 7.

- Section 7- Conclusions could be further elaborated by gathering nice comments previously included in the paper and by precising better some aspects: It is shown that by acting on turbine 1, turbine 3 remains unaffected.

  The observation that "most of the gain [is] coming from the first downstream turbine" has been added to the conclusions.

- It is shown that, for a given mean wind speed, the change in the power gain mostly depends on the amplitude of the DIC and not on the frequency. Would it be any dependence on the mean wind speed? The experiments have examined the effect of DIC under different TI conditions. It would also be interesting to see in the future the effect under different mean wind speed conditions.

The authors absolutely agree with the referee that investigating the effect of different mean wind speed conditions would be very interesting. It would for example be very informative to check whether DIC would also work with above-rated wind speeds, when the pitch angle is already varied to ensure constant power output. Therefore, this suggestion has been added to the future research opportunities in Section 7.

- Page 15, Line 17 to Page 16, Line 1: In all, it can be concluded that the dynamic induction control approach shows great promise, as now both simulations and scaled experiments show that it is possible to achieve a power gain. However, significant differences are found between simulation and experiments, which still need to be addressed. The conclusion included does not apply to the presented simulation results, which consist in the simulation of one single turbine, mainly for loading evaluation. These simulations dont provide insights into the behavior and power gain at farm level. Equally, it is not clear which are the significant differences between simulation and experiments this statement makes reference to.

  This comment is similar to comment 16 of Prof. Meyers, so the response is also similar. This comment refers to differences between the results found in simulations executed by Munters and Meyers, and the wind tunnel results presented in this paper. This is now clarified more explicitly. Most notably, the optimal frequency and amplitude of excitation is found to be slightly higher and lower respectively. To name these differences "significant" might be a bit too definite, so this was changed to "some minor differences". Furthermore, the aforementioned differences are now explicitly named in a prior paragraph of the conclusions.

- Is there any hypothesis on why the increase in the DIC amplitude provokes such decrease in the final power gain?

  As already discussed in Section 6, the power loss is caused by a very significant drop in power production of the excited turbine with higher DIC amplitudes, for which downstream machines cannot fully compensate. A possible explanation for this could be a slight rotor imbalance which was present in the $G1$ models, which causes significant vibrations on the excited turbine for higher amplitudes of excitation. This explanation has been added to both Section 6 (results) and Section 7 (Conclusions).

- For practical application of the technology, taking into account that DIC is intended for region II -among others-, have you considered the possible risk of stall when applying a periodic pitch variation of several degrees around fine pitch? The value of 2 degrees used in simulations (section 5) could prove to be relevant.

Stall is not something we have looked into as of yet, although we are of course aware of this risk. However, this did not prove to be a problem in the scaled experiments, as quite extreme pitch variations (up to $\pm 5°$) were used without stall issues. Investigating the risk of stall on full scale machines, although of course very interesting, is out of the scope of this research.

- The lowest tested amplitude for DIC has proved to be the best one. So, one question that arises is whether further decrease in the amplitude would lead to even better results. It would be interesting to determine in the future which is the minimum "A" that provides the maximum power gain.

The authors fully agree with this observation. For this reason, it is also clearly mentioned in the conclusions that further experiments are necessary to determine the full possibilities of periodic DIC.

- In the wind tunnel experiments it has been possible to measure the thrust coefficient thanks to the knowledge about the wind conditions. This has allowed the determination by trial and error of the pitch variation in order to provide a thrust coefficient (amplitude, frequency) matching the desired one. How would this technology be applicable in real wind turbines where such detail of information about wind conditions is not so easily and precisely available?

In the experiments presented here, a excitation of the collective pitch was used to create a certain desired thrust coefficient. Assuming the optimal settings are independent of the wind speed (which is yet to be investigated), the optimal pitch excitation could simply be used without knowledge on the wind conditions. However, a far more interesting solution, which is also mentioned in the future research opportunities, is to develop a closed-loop dynamic induction control algorithm, including an engineering model or observer to estimate the wind conditions. This controller would then determine the optimal DIC settings and would be able to adapt to changing wind conditions based on the latest measurements of, for example, the turbine power production.

- For the sake of clarity and reproducibility: It would be advisable to indicate upfront from the very beginning of the paper that it focuses on below rated conditions and excitation of collective pitch angle. Also, to leave an explanatory comment about induction as in-wake speed deficit.

Both the below-rated testing conditions and the induction definition have been included in the introduction.

- Table 1: Missing frequency units in last row (Frequency of excitation in St). Its understood that it is Hz, but better to leave it explicit.

As mentioned in the text, the Strouhal number $St$ is actually dimensionless. For clarity, "[-]" was added after $St$ to note this dimensionlessness.

- Table 2: Please make coherent the denomination for the amplitude variable A (third column in the table) with the description in the table caption (CT,DIC).

  Due to a different comment from another referee, the caption of Table 2 has been modified. The denominations are now all coherent.

- Page 7, Line 18: It could be added as examined load the hub torsional moment, taking into account that these results are presented in Table 3.

  The mention of the hub torsional moment has been added here.

- Page 8, Line 9: It could be added mean therefore indicating mean hub wind speed of

  The addition of the word "mean" has been implemented as requested.

- Figure 7 and Figure 8, caption: It could be added mean therefore indicating mean wind speed

  The addition of the word "mean" has been implemented as requested.

- Table 3. The table caption would be clearer if it is indicated that the percentages refer to improvement with respect to baseline. Equally, it is indicated AEP in the caption, although the values are not included in the table. The percentage of variation of power with respect to baseline is of great interest, in order to compare the order of magnitude with the results of turbine 1 in the wind tunnel experiments. So, it would be advisable to introduce such information, not only in terms of AEP, but also through a figure of comparison with baseline, for example power time plot corresponding to Figure 5.

  The caption has been augmented to include that the results are given with respect to the baseline. AEP values of the excited turbine have been included. To accomodate the desire of the referee, a figure of the AEP over time has also been added to the paper.

- Section 6. It would be advisable to indicate the layout of the wind farm tested in the wind tunnel, either through written explanation or through a descriptive figure.

  The authors completely agree that such a figure was missing from the paper. In Section 4, explaining the wind tunnel setup, the requested figure showing the layout of the wind farm in the wind tunnel has been added.

- Table 4, caption: Caption could be clearer by making reference to baseline: An overview of the total power increase with respect to baseline by applying

  As requested, the text "with respect to the baseline case" has been added in the caption of Table 4.

- Table 4 and Table 5: It would be advisable to indicate the frequency units (first row).

  The requested frequency units have been added as requested.

- Page 11, Line 5: When mentioning the change of $+2\%$ in blade root loads, it would be advisable to specify flapwise. Equally, when mentioning the negligible impact found in edge-wise and in the hub, it would be clearer to mention the respective percentages, since for edgewise, its only 0.4%, but for the hub it accounts for 1% to 2%.

  All suggested additions have been implemented.

- The discussion of load results is mainly done for St = 0.4 and St = 0.5, while the best fit for experiments is provided by St = 0.33 (low TI) and St = 0.29. Which would be the correspondence between the St results in the scaled tests and those for a full-scale model such as the one simulated in CP-LAMBDA?

  It is hard to say how the optimal Strouhal number scales with the turbine size. The full-sized turbines used by Munters and Meyers find an optimum of $St = 0.25$, and the Strouhal number does scale for rotor size, so it could be argued that the optimal Strouhal number is (relatively) independent on the rotor size. This is something that could still be investigated in the future. The analysis done here focusses on the possible load effects for different Strouhal numbers, without arguing which of these would be optimal for power production in this case. The discussion of the results has been changed to include $St = 0.3$.

- Page 11, Line 18: When making reference to the experiments with different amplitudes on a sinusoidal input, it would be convenient to introduce the reference to Table 2. Equally, it could be helpful to indicate again that the sinusoidal input is applied to the collective pitch, which is the range of variation of the pitch angle, and which correspondence this would have with the pitch angle in a full-scale wind turbine.

  The requested reference to Table 2 has been added. The authors feel that this refence suffices as all the information requested by the referee can be found in this table. By focussing on the amplitude of the $C_T$-excitation, the authors also feel that a notion on scalability of the pitch amplitude is unnecessary: this might differ per turbine, but can easily be calculated with the required $C_T$-$\beta$-tables.

- Page 13, Line 3. In the same way that it is indicated explicitly for low TI experiments (Page 11, Line 17), it would be nice to indicate the approximate value of TI applied in the high TI experiments.

  As requested, the high-TI value (10%) has been added here.

- Page 13, Line 6. For higher clarity, it could be indicated to which production it makes reference the sentence. It is understood that it refers to: the baseline power production of this turbine is already slightly lower than in low TI conditions.

  The referee is correct in his assumption. For clarity, the suggested addition has been made.

- Page 14, Line 8: For the sake of clarity, it would be advisable to introduce again the reference Schreiber et al. (2017), which was already indicated in Page 4.

  The requested reference has been added here.

- Page 3, line 8: were instead of where

  This erratum has been corrected.

- Table 1 The frequencies of excitation in St indicated for the aeroelastic simulations Between 0.3 and 0.5 dont match the range of frequencies of DIC stated in Section 5, Page 8, where it is stated that this frequency varies from 0.00952 Hz to 0.0595 Hz. Equally, the frequencies indicated for the experiments [0.09-0.41] dont match the frequencies included in Table 4 and Table 5 [0.5-2.3].

  The referee seems to confuse two different units here. In general, the frequency of excitation is expressed with the dimensionless Strouhal number, as defined in Section 2. This unit is also used in Table 1, so the values given here are dimensionless, not in Hertz. They do in fact match with the values of $St$ given in Tables 4 and 5, as well as the values of $St$ mentioned on page 8.
  To prevent such confusion in a future version of the manuscript, the word "frequency" has been removed from Table 1, which now reads "Strouhal number $St$ of excitation [-]". Table 4 and 5 already contained both the frequency in Hertz as well as the Strouhal number, but units have been added to clarify the difference. Hopefully this removes the confusion and helps the referee understand the implemented control signals.

- Page 6, line 15: kHz instead of kH

  This erratum has been corrected.

- Figure 5, xlabel: It would be preferable to indicate time units in accordance to the symbol stated by the International System of Units: s

  The units have been changes from "sec" to "s".

- Figure 7 and Figure 8, xlabel: It could be introduced a space between Wind Speed and the unit [m/s]

  A space has been added before the unit.

- Page 11, Line 1: According to SI unit rules and style conventions, unit should not be italic m/s.

  The unit is no longer displayed in italic.
- Page 11, Line 3: In accordance to style convention, there should be a space between the number and unit 15 m/s

  A space has been added.
- Page 11, Line 22: It seems that the verb is missing in the sentence: the power is divided

  This is corrected as suggested by the referee.
- Figure 9, Caption: The reference in the figure legend and caption should be coherent between CT and CT.

  As a response to a different comment, the legend and caption of this figure has already been changed. The amplitude is now given by the variable $A$ in both the legend and caption.
- Figure 11, legend: It seems that baseline would fit better than "benchmark", also keeping coherence with previous figures such as Figure 9.

  This has been corrected.
- Page 14, Line 2: It seems that the sentence However, since the power gain at turbine 3 is slightly lower, the total power is also lower than in the baseline case would indeed make reference to turbine 2, according to the figures.

  The referee is right in his assumption, and this has been corrected.
- Page 15, Line 15: To be corrected weighted instead of weighed.

  This has been corrected.
- It would be preferable to specify the increase of the weighted DEL with respect to baseline. Equally, the values of DEL included could be misleading without specifying which load they make reference to. Indeed, the 0.3-0.4% refers to blade root edgewise, which is the least affected by DIC.

  The addition "with respect to the baseline case" has been added, as well as the notion that these number refer to the blade root edgewise loads.

**Response to comments of Anonymous Referee #4**

- Dear authors, Thank you very much for submitting the paper to the WES journal. It was nice reading the paper and it is of high quality. Altogether a lot of relevant work is presented and it gives a significant contribution to the community. The paper follows a clear structure and gives a lot of background information that helps to understand the tasks that have been performed. Altogether I recommend the publication with the consideration of the following minor corrections and the comments of the other reviews.

  The authors would like to thank the referee for the compliments, as well as for the constructive feedback in improving the quality of the paper.

- Abstract: Please introduce the idea of induction control before naming it and extend the abstract a litte more. This would help people being not familiar with the topic to unterstand the content of the paper.

  The abstract has extended: it now includes a (very general) introduction into wind farm control as well as in induction control. The additions made are as follows: *As wind turbines in a wind farm interact with each other, a control problem arises that has been extensively studied in literature: how can we optimize the power production of a wind farm as a whole. A traditional approach is to this problem is called induction control, in which the induction factor, i.e. the in-wake wind speed deficit, of a turbine is lowered such that downstream turbines can increase their power capture.*

- Figure 1: Please explain the figure in more detail in the caption. This figure basically presents the whole concept and needs therefore more explanation.

  A much more elaborate caption has been added to this figure, to better explain the concepts shown here.

- p. 2 l.4: you say DTU 5 MW turbine: NREL 5 MW turbine

  This erratum has been corrected.

- Table 1: Munters et. al.

  This erratum has been corrected.

- Table 1: please first introduce beta and cT before having the table. I know that latex is placing it like this, but moving it to the next page is preferable.

  The paragraph introducing these variables is moved forward, such that it precedes the table, as well as the first mention of the table.

- Page 4: yaw control: to me wake steering is more familiar than yaw control. Maybe you need to add both or replace it

  Both "yaw control" and "wake redirection control" are now explicitly mentioned here.

- Figure 7-12: the style of the labels differ to the previous plots,

  The difference in style has been removed: all labels are now in "normal" letter style.

- Figure 7, 8: a space before unit (As mentioned in caption Fig. 8)

  The space before the units has been added.

- Conclusions: p.16 l.1: please again name the differences in the conclusions

  The differences, namely a slightly different optimal Strouhal number $St$ and amplitude $A$, are now explicitly mentioned again in the conclusions.

- Acknowledgements: program: programme

  This erratum has been corrected.

---

## Editor Decision (ED1)

Abstract
- Second sentence overly complicated

Introduction
- Typo in third sentence "with induction the in-wake speed deficit"
- Explanation of Figure 1 in the text could be better
  - In the caption, a rough indication of what the arbitrary induction settings mean as well as the control methods used for affecting those induction settings would be better.
  - The term greedy control is used in the figure caption without explanation
- "the optimal dynamic control inputs…" which ref? also can you be more specific about how the induction factor is varied over time from the Meyers work 2015, 2017?
- Be careful when using optimal terminology, a grid search as described would not necessarily yield an optimum.
- The Stouhal number is introduced on first use without definition (which is in the following section). Recommend leaving out the numeric specifics in the intro until they can be introduced in more detail in the next section. Instead, focus on a better description of Figure 1 and the qualitative aspects of what is going on… it could be much better explained to help the reader develop the intuition before jumping to the numerics.

Control strategy
- What do you mean time constraints? Access to the tunnel?
- Table 1: Differences in the table could be better explained. Strouhal ranges overlap but are quite different. Can you show this better? It's a bit confusing why there are differences and what was done to address them.  In table mixing items related to pitch setting and Ct, these should be separated out and where there is a lack of information in a specific cell, put N/A or dashes. Avoid nonspecific language like "between", use either discrete list of numbers or a distribution function – whatever is applicable
- The use of collective pitch control is only first introduced in line 24 and no other mention of induction control methods are introduced. There should be some better explanation of the different methods and why CPC was used
- Yaw control first introduced in line 31 without explanation- why does it pop up here? Why is it worth comparing DIC with yaw control? Again, this should be explained. It should also be mentioned that this will be done in the introduction. Overall the paper is lacking clear explanation of various control strategy options and implementation methods.

Simulation environment
- Why the NREL 5 MW? What limitations are there for this model?

Experimental setup
- This section is well written and answers some of the previous questions. Discussion on Ct approach in control system starting at line 14 should be brought forward to intro/control strategy sections

Simulation results

- Better justify selection of load sensors
- Weak justification of performing the range of 3 to 25 m/s… the effectiveness of the technique will taper off above rated. Going to 25 m/s is not well justified in sentence 7
- Better explain the sentence on lack of DIC input frequency / turbine vibratory modes… for instance give example of typical turbine frequencies (or those of 5 MW) to show that they are well outside the range of DIC frequency
- Figures 7-9 and table 3: using the entire range of operational wind speeds from 3 to 25 m/s is overestimating the impact of the control technique on the loads… it is likely that the technique will not do much in terms of power production improvement beyond something like 15 m/s. it is not clear that the Weibull approach fully counteracts this
- The statement that it is the worst case possible contradicts the use of the Weibull to weight the loads…

Experimental results
- Bold in table 4 not described in text or caption
- 6.2 controller comparison – explanation of control strategies should appear up in the introduction / control strategy sections

Overall
- This paper would have been better split into two papers with more comprehensive focus on each of the power production improvement possibilities and comparison to experiments and a second paper on loads implications. Each individual area feels incomplete – particularly the loads analysis section.

Referee comments
- A few comments have been marked XXX, these should be addressed.

---

## Author Response (AR2)

| | |
|---|---|
| Date | December 19, 2019 |
| Our reference | WES-2019-50 |
| Contact person | J.A. Frederik |
| Telephone/fax | +31 (0)15 27 85623 / n/a |
| E-mail | J.A.Frederik@TUDelft.nl |
| Subject | Author's response |

**Delft University of Technology**

Delft Center for Systems and Control

Address
Mekelweg 2 (3ME building)
2628 CD Delft
The Netherlands

www.dcsc.tudelft.nl

dr. Katherine Dykes
*Wind Energy Science Discussion*

Dear dr. Dykes,

First of all, the authors would like to thank the Associate Editor for her positive and constructive feedback. We believe that the comments have helped us to further improve the quality of the paper. In our attempt to account for the comments, we have revised different aspects of the paper. The objective of this document is to respond to the points raised by the Associate Editor and to provide a detailed overview of the changes made to the paper. In the subsequent section, we will respond to the Associate Editor Decision report. At the end of this document, the changes made to the manuscript are documented.

Yours sincerely,

On behalve of the authors,

Joeri Frederik

Enclosure(s): Response to Associate Editor Decision

**Response to Associate Editor Decision**

**Abstract**

- Second sentence overly complicated
  This sentence has been simplified to: *A traditional approach to this problem is called induction control, in which the power capture of an upstream turbine is lowered for the benefit of downstream machines.*

**Introduction**

- Typo in third sentence "with induction the in-wake speed deficit"
  This typo is corrected to: *with induction defined as the in-wake speed deficit.*

- Explanation of Figure 1 in the text could be better
  - In the caption, a rough indication of what the arbitrary induction settings mean as well as the control methods used for affecting those induction settings would be better.
    Indications of approximate induction settings have been added to the caption, including a description of possible control methods: *The orange ($a \approx 0.3$) and yellow ($a \approx 0.25$) lines depict arbitrary static derating settings that can be achieved by changing either the generator torque or the collective pitch angles of the turbine.*
  - The term greedy control is used in the figure caption without explanation
    The term greedy control is now explained in the caption: *The force $F_T$ exerted on the wind is shown for different induction settings $a$, where red depicts "greedy" settings that result in optimal single turbine power capture ($a = 1/3$).*

- "the optimal dynamic control inputs..." which ref? also can you be more specific about how the induction factor is varied over time from the Meyers work 2015, 2017?
  An additional reference to Munters and Meyers (2017) is added to clarify the reference. Furthermore, the control approach of this work is now elaborated in the text: *The thrust coefficient $C'_T$ of each turbine is used as the control input. This input is only constraint by different wind turbine response times $\tau$ and maximum allowable thrust coefficient settings $C'^{max}_T$, resulting in non-smooth control signals.*

- Be careful when using optimal terminology, a grid search as described would not necessarily yield an optimum.
  The term "optimal" is now dropped in this specific sentence. It now reads: *A grid search with different amplitudes and frequencies is performed to find the periodic dynamic signal that results in the maximum energy extraction in a high-fidelity simulation environment.*

- The Stouhal number is introduced on first use without definition (which is in the following section). Recommend leaving out the numeric specifics in the intro until they can be introduced in more detail in the next section. Instead, focus on a better description of Figure 1 and the qualitative aspects of what is going on... it could be much better explained to help the reader develop the intuition before jumping to the numerics.
  The authors agree that it is better to leave out the Strouhal number at this point of the paper. This sentence is therefore removed from the paper.

Control strategy

- What do you mean time constraints? Access to the tunnel?
  Indeed, this means access to the tunnel. This phrase has been changed to clarify this: *The amplitude and frequency ranges were slightly reduced due to limits on the available time in the wind tunnel.*

- Table 1: Differences in the table could be better explained. Strouhal ranges overlap but are quite different. Can you show this better? It's a bit confusing why there are differences and what was done to address them. In table mixing items related to pitch setting and Ct, these should be separated out and where there is a lack of information in a specific cell, put N/A or dashes. Avoid nonspecific language like "between", use either discrete list of numbers or a distribution function – whatever is applicable
  The text preceding the table now explains how the specific control settings were chosen. Furthermore, the amplitude of pitch and $C_T'$ are now given in individual rows of the table as requested. Finally, the range and the number of data points are given to define the different experiments. The authors feel that a discrete list of numbers is redundant at this point, as these can be found in the Tables showing the results (Tables 3, 4 and 5).

- The use of collective pitch control is only first introduced in line 24 and no other mention of induction control methods are introduced. There should be some better explanation of the different methods and why CPC was used

  The reference to the table is placed further back, such that the collective pitch approach is mentioned before the table. The induction control approaches are also further elaborated: *Finally, a method should be found to vary the thrust coefficient of a real (scaled) wind turbine. The thrust coefficient can be manipulated by varying either the collective pitch angle or the generator torque of the turbine. Of these two, the former approach is the most straightforward and easy to implement. Therefore, the collective pitch angle $\beta$ of the upstream model was excited periodically.*

- Yaw control first introduced in line 31 without explanation- why does it pop up here? Why is it worth comparing DIC with yaw control? Again, this should be explained. It should also be mentioned that this will be done in the introduction. Overall the paper is lacking clear explanation of various control strategy options and implementation methods.

  The comparison cases that will be used in Section 6, which includes yaw control, are now described more extensively in Section 2. Furthermore, it is explained more explicitly why these comparison cases are used: *Finally, the performance of periodic DIC as a wind farm power maximization strategy will be evaluated. To achieve this, a comparison will be made with wind farm power maximization approaches that have already been investigated more extensively in literature.* This comparison was already mentioned in the introduction, but is made more explicit: *As comparison cases, static induction control and wake redirection control (Fleming et al, 2014), where upstream turbines are yawed with respect to the wind direction to redirect the wake away from downstream machines, are implemented in the wind tunnel.* Hence we expect to have removed the suggested lack of clear control approach explanation.

Simulation environment

- Why the NREL 5 MW? What limitations are there for this model?

  NREL5MW is a widely used wind turbine, well known in literature and still representative of modern operating wind turbines. The idea of this numerical task is to preliminary investigate the effect of this wind farm controller on the fatigue loads, with the main goal of understanding how this technology may be applied also on existing wind farm. A full load analysis is out of the scope of this paper but is subject of ongoing research. Thank you again for this comment, we took the opportunity to highlight better this aspect at the beginning of this section.

Experimental setup

- This section is well written and answers some of the previous questions. Discussion on Ct approach in control system starting at line 14 should be brought forward to intro/control strategy sections

  The authors thank the editor for the compliment. As requested, the discussion regarding the $C_T$-approach is brought forward to the control strategy section.

Simulation results

- Better justify selection of load sensors

  Sensors have been placed in the main sub-components of the wind turbine, such as blade root, tower top and base, hub and nacelle. Moreover, we have all the controller data. In the paper we have showed the ones that may be influenced by the controller.

- Weak justification of performing the range of 3 to 25 m/s... the effectiveness of the technique will taper off above rated. Going to 25 m/s is not well justified in sentence 7

  The DIC is useful for low wind speed values (i.e. region I, II and the begining of region III). At higher wind speed the wake is sufficiently energized, so there is no need for mixing. In the following figures (8/9), we wanted to show what would happen with an always-on DIC, i.e. the worst scenario, where the DIC is always switched on for any wind speed and wind direction. We have updated the text in the paper to highlight better this point.

- Better explain the sentence on lack of DIC input frequency / turbine vibratory modes... for instance give example of typical turbine frequencies (or those of 5 MW) to show that they are well outside the range of DIC frequency

  Thank you again for this point. We have included in the text the range of the most important aeroelastic frequencies to highlight better the separation between the latter and the DIC frequencies.

- Figures 7-9 and table 3: using the entire range of operational wind speeds from 3 to 25 m/s is overestimating the impact of the control technique on the loads. . . it is likely that the technique will not do much in terms of power production improvement beyond something like 15 m/s. it is not clear that the Weibull approach fully counteracts this
  As explained in the previous point, here we considered the DIC on in the full range of wind speed values, as seen in pictures 8-9 (not 7 which shows a PSD at 15m/s) where at high wind speed the baseline DELs are different wrt the DIC ones. The power production improvement cannot be seen here because the analysis refers to the upstream wind turbine only, where the DIC is operating. Table 3 shows, differently from the previous figures, the Weibull-weighted DEL where we have supposed to switch off the DIC for wind values higher than 15m/s. Consistently we have computed the AEP variation. Again we updated the text to better explain these results and in the table 3 label.

- The statement that it is the worst case possible contradicts the use of the Weibull to weight the loads. . .
  As explained in the previous point, here we considered the DIC on in the full range of wind speed values, as seen in pictures 8-9 (not 7 which shows a PSD at 15m/s) where at high wind speed the baseline DELs are different with respect to the DIC ones. The power production improvement cannot be seen here because the analysis refers to the upstream wind turbine only, where the DIC is operating. Table 3 shows, differently from the previous figures, the Weibull-weighted DEL where we have supposed to switch off the DIC for wind values higher than 15m/s. Consistently we have computed the AEP variation. Again the text and the caption of Table 3 have been updated to better explain these results.

Experimental results

- Bold in table 4 not described in text or caption
  An explanation is now added to the caption: *In bold are the experiments that lead to the highest power capture for each amplitude, showing an optimum around $St = 0.28$.*

- 6.2 controller comparison – explanation of control strategies should appear up in the introduction / control strategy sections
  As discussed in response to one of the previous comments, the explanation of the control strategies was moved to Section 2 (Control Strategy).

Overall

- This paper would have been better split into two papers with more comprehensive focus on each of the power production improvement possibilities and comparison to experiments and a second paper on loads implications. Each individual area feels incomplete – particularly the loads analysis section.

  Although the authors respect the opinion of the Associate Editor and are thankful for her constructive feedback, they disagree that the research presented here is incomplete. It is mentioned on several occasions in the paper that the main goal of the load investigation is to understand how this technology may affect existing wind turbines and wind farms. Furthermore, the authors feel like this comment would have been more useful in an earlier stage of the review process. As the Associate Editor mentions herself, additional work is at this stage not recommended. Finally, the authors would like to point out that none of the original four referees made any comments regarding potential incompleteness of the research presented in this paper.

Referee comments

- A few comments have been marked XXX, these should be addressed.

  The Associate Editor refers to the following comments from the referees in the interactive discussion:

  - Figure 6: This is a really useful view into the loading impacts. Is there a reference for Weibull-weighted DELs? A nice idea, are they used often?

    The single DEL is obtained by reducing the time histories of stochastic loads obtained in 10min simulations (one for each mean wind speed) in turbulent wind into simple cyclic loads by the well-known rain-flow analysis. This gives the amplitude of an equivalent (in term of fatigue damage) periodic signal for that wind speed. The Weibull-weighting procedure hence account for the loads in all the wind speed spectra (also considering the parked conditions). This procedure is widely used to extract a simple, but accurate, indication of the cumulative fatigue load in the Industry and in the Academia. This is then used to compare the effects of different controllers and/or wind characteristics (turbulent intensity, seeds, etc.). To our knowledge, one of the first paper presenting this approach is "Carlo L. Bottasso and Alessandro Croce and C.E.D. Riboldi and Y. Nam, Multi-Layer Control Architecture for the Reduction of Deterministic and Non-Deterministic Loads on Wind Turbines, Renewable Energy, Volume 51, March 2013, Pages 159-169 , 2013", even if some of the authors already used this analysis in industrial projects to estimate the fatigue on wind turbine sub-components (bearings, actuators, etc.). Hence, it is hard to figure out who was the first that introduced this concept and to provide a reference. In any case, since we defined the Weibull-weighted DEL through Eq. (2), the text results complete and does not need modifications.

- 5. page 4: A region I-1/2 with constant rotor speed equal to 6 rpm extends from the cut-in speed of 4 m/s to 7 m/s. Im a bit surprised by this please double check. As far as I remember, in region 1.5 the rotor speed is increasing, and not constant.

  The referee is right, the above description was referred to another wind turbine. Furthermore, since the reference of NREL 5MW wind turbine has been provided, and the reader may find all data including a detailed description of the control law, we decided to remove that sentence, which does not provide any essential information.

[revised manuscript text omitted]

To verify the validity of the periodic dynamic induction approach for fast wake recovery in a wind farm, a number of wind tunnel experiments in both low and high Turbulence Intensity (TI) conditions are executed. All experiments are executed at a below-rated wind speed, i.e., in operating region II. The effect of varying the amplitude and frequency of the signals is studied, and the performance of this approach is compared with  state-of-the-art wind farm power maximization control strategies. As comparison cases, static induction control and wake redirection control (Fleming et al., 2014), where upstream turbines are yawed with respect to the wind direction to redirect the wake away from downstream machines, are implemented in the wind tunnel. A positive result in these experiments would be an important step towards proving the validity of this approach in real wind farms.

The structure of this paper will be as follows: in Section 2, the DIC strategy will be explained. Sections 3 and 4 will elaborate on the simulation environment and the experimental setup, respectively. In Section 5, the simulation results will be presented, followed by the experimental results obtained in the wind tunnel in Section 6. Finally, the conclusions will be drawn in Section 7.

**2    Control Strategy**

In this section, the strategy behind dynamic induction control will be discussed shortly. As mentioned in the introduction, the approach presented in Munters and Meyers (2018) is used as a basis for this paper: the thrust force of the upstream wind turbine is excited to induce wake mixing, in order for downstream turbines to increase their power capture. It is shown that the amplitude and frequency of a sinusoid determine the overall power production. The optimum found in here is a Strouhal number of $St = 0.25$, with an amplitude of the disk-based thrust coefficient $C'_T = 1.5$. The Strouhal number is defined as $St = fD/U_\infty$ for a given frequency $f$, rotor diameter $D$ and inflow velocity $U_\infty$, while $C'_T = 4a/(1 - a)$, with $a$ the axial induction factor (Goit and Meyers, 2015). This disk-based thrust coefficient relates to the thrust coefficient $C_T$ as $C_T = C'_T(1 - a)^2$. For the `G1` models and an inflow velocity of 5.65 m/s, this Strouhal number would result in an excitation frequency of approximately 1.3 Hz.

However, there are some fundamental differences between Munters and Meyers (2018). First of all, due to the size of the wind tunnel (see Section 4), a 3-turbine wind farm is the deepest possible array configuration. The amplitude and frequency ranges were slightly reduced due to  limits on the available time in the wind tunnel. Furthermore, the number of experiments executed in this paper is slightly lower. The amplitudes and frequencies for the wind tunnel experiments are chosen such that sufficient data points can be investigated around the optimum found in Munters and Meyers (2018). For the aero-elastic simulations, three different frequency points are evaluated to demonstrate the effect on the turbine loads. Finally, a method should be found to vary the thrust coefficient of a real (scaled) wind turbine. The thrust coefficient can be manipulated

[Figure]

**Figure 2.** Values of $C_T$ for different types of input signals, created using a look-up table of the `G1` turbine model. The thrust coefficient is shown for three different sinusoidal excitations: on $C_T$, on $C_T'$ and on the collective pitch angle $\beta$, tuned such that the amplitude of $C_T'$ is 1.5. The dashed line shows the steady-state optimal $C_T$.

by varying either the collective pitch angle or the generator torque of the turbine. Of these two, the former approach is the most straightforward and easy to implement. Therefore, the collective pitch angle $\beta$ of the upstream model was excited periodically. This results in a slightly different thrust signal, as shown in Figure 2, but simulations show that the difference in output for these input signals is limited. All these differences are summarized in Table 1

**Table 1.** Differences between the approach in Munters and Meyers (2018) and both the simulations and wind tunnel experiments pre-sented in this paper.  The number of experiments executed here is slightly lower. As a result, choices are made with regards to the excitation amplitudes and frequencies that have been investigated.

| | Munters  | Simulations | Experiments |
|---|---|---|---|
| **Layout** | 4 turbines in a row | Single turbine | 3 turbines in a row |
| **Environment** | LES code | Aero-elastic code | Wind tunnel experiments |
| **Control input** | Sinusoid on $C_T'$ | Sinusoid on $\beta$ | Sinusoid on $\beta$ |
| **Amplitude of pitch excitation** |  N/A |  2 |  1.7, 2.8, 5 |
|  **Amplitude of $C_T'$ excitation** |  0.5, 1, 1.5, 2 |  1 |  1, 1.5, 2 |
| **Number of frequency data points** | 12 | 3 | 8 |
| **Frequency range in $St$ [-]** | $[0.05, 0.6]$ | $[0.3, 0.5]$ | $[0.09, 0.41]$ |

**Table 2.** Average $\bar{C}_T$ and amplitude $A_{C_T}$ of the three different thrust coefficient oscillations whose results are discussed in Section 6, as well as the mean pitch angle average $\bar{\beta}$ and amplitude $A_\beta$ used to achieve these signals. Note that, as explained in Section 2, these collective pitch settings are not identical for different frequencies. Instead, they are tuned such that the mean and amplitude of $C_T$ as given below are followed as accurately as possible.

| Amplitude $C_T'$ | $\bar{C}_T$ [-] | $A_{C_T}$ [-] | $\bar{\beta}$ [deg] | $A_\beta$ [deg] |
|---|---|---|---|---|
| $A = 1$ | 0.8 | 0.17 | 0.7 | 1.7 |
| $A = 1.5$ | 0.7 | 0.3 | 1.8 | 2.8 |
| $A = 2$ | 0.5 | 0.5 | 4 | 5 |

~~Since the internal torque controller of the G1 model is also active, the amplitudes and offsets of the pitch signals are tuned manually such that the resulting thrust coefficient matches the desired thrust coefficient in amplitude and frequency. To achieve this, the thrust force on the turbine is measured, which, together with knowledge about the wind conditions , is used to calculate the thrust coefficient over time.~~ For the tests performed within the research described in this paper, the standard power controller was augmented in order to enable the rotor thrust coefficients following a specific sine wave function. However, there is not a unique way of achieving this goal, since a specific thrust coefficient $C_T(\lambda, \beta)$ can be obtained by operating at different combinations of tip-speed-ratio $\lambda$ and blade pitch $\beta$. In turn, the tip speed ratio can be varied either by changing the reference followed by the generator torque or changing the blade pitch. In this paper, a strategy that only changes the blade collective pitch is adopted. The implementation of this strategy simply requires changing the collective fine pitch at which the model blades are set when the machine operates in partial load conditions (region II). The fine pitch was tuned experimentally, by means of a trial and error procedure conducted with a stand-alone model, to achieving the desired mean $\bar{C}_T$ and amplitude $A$ as reported in Table 2. The effects of these control actions in terms of impacts on the power output of the 3-turbine wind farm will be discussed in Section 6.

Finally, the performance of periodic DIC as a wind farm power maximization strategy will be evaluated. To achieve this, a comparison will be made with wind farm  power maximization approaches that have already been investigated more extensively in literature:

- *Greedy control*: all turbines operate at their individual optimum, disregarding wake interaction between turbines. This means that all turbine have an induction factor of $a = 1/3$ (or a thrust coefficient of $C_T = 8/9$ or $C_T' = 2$) and a yaw angle of 0 degrees with respect to the wind direction.

- *Static induction control* (also called derating control): the induction settings of upstream turbines are manipulated such that the wind farm power capture can be maximized. In this paper, the induction factor is controlled by means of the collective pitch angles of the (upstream) turbines, although using the generator torque is also an option. This

strategy has been a popular research topic in recent years, and has shown both promising (Marden et al., 2013; Gebraad et al., 2013) and inconclusive (Campagnolo et al., 2016a; Nilsson et al., 2015; Annoni et al., 2016) results.

- *Yaw control* (also called wake redirection control). : upstream turbines are yawed with respect to the wind direction such that the wake is steered away from downstream machines. For this approach, the control inputs are the yaw angles of the (upstream) turbines with respect to the wind. Yaw control has been demonstrated to effectively increase the wind farm power capture in wind tunnel experiments (Campagnolo et al., 2016c) and full-scale experiments (Fleming et al., 2017; Howland et al., 2019).

[revised manuscript text omitted]

IEC 61400-1 Ed.3.: Wind Turbines — Part 1: Design requirements, Tech. rep., Garrad Hassan and Partners Ltd, St Vincent's Works, Silver-thorne Lane, Bristol BS2 0QD, UK, 2004.

Munters, W. and Meyers, J.: An optimal control framework for dynamic induction control of wind farms and their interaction with the atmospheric boundary layer, Phil. Trans. R. Soc. A, 375, 20160 100, 2017.

5  Munters, W. and Meyers, J.: Towards practical dynamic induction control of wind farms: analysis of optimally controlled wind-farm boundary layers and sinusoidal induction control of first-row turbines, Wind Energy Science, 3, 409–425, 2018.

Nilsson, K., Ivanell, S., Hansen, K. S., Mikkelsen, R., Sørensen, J. N., Breton, S.-P., and Henningson, D.: Large-eddy simulations of the Lillgrund wind farm, Wind Energy, 18, 449–467, 2015.

Schreiber, J., Nanos, E., Campagnolo, F., and Bottasso, C. L.: Verification and calibration of a reduced order wind farm model by wind tunnel

10  experiments, in: Journal of Physics: Conference Series, vol. 854, p. 012041, IOP Publishing, 2017.

Westergaard, C. H.: A method for improving large array wind park powre performance through active wake manipulation reducing shadow effects, 2013.

---

## Author Response (AR3)

Date          January 21, 2020
Our reference WES-2019-50
Contact person J.A. Frederik
Telephone/fax +31 (0)15 27 85623 / n/a
E-mail        J.A.Frederik@TUDelft.nl
Subject       Author's response

**Delft University of Technology**

Delft Center for Systems and Control

Address
Mekelweg 2 (3ME building)
2628 CD Delft
The Netherlands

www.dcsc.tudelft.nl

dr. Katherine Dykes
*Wind Energy Science Discussion*

Dear dr. Dykes,

The authors are delighted that the paper is accepted for publishing in Wind Energy Science. We would like to sincerely thank you, as well as the reviewers, for the feedback, which has helped in producing the best possible journal paper.

For the final version of the paper, only a small number of typos has been corrected. Furthermore, some notations have changed and a couple of complementary sections are added at the end of the paper to comply to the WES manuscript composition guidelines.

Yours sincerely,

On behalve of the authors,

Joeri Frederik

Enclosure(s):  Changes made to the manuscript

[revised manuscript text omitted]

To verify the validity of the periodic dynamic induction approach for fast wake recovery in a wind farm, a number of wind tunnel experiments in both low and high Turbulence Intensity (TI) conditions are executed. All experiments are executed at a below-rated wind speed, i.e., in operating region II. The effect of varying the amplitude and frequency of the signals is studied, and the performance of this approach is compared with state-of-the-art wind farm power maximization control strategies. As comparison cases, static induction control and wake redirection control (Fleming et al., 2014), where upstream turbines are yawed with respect to the wind direction to redirect the wake away from downstream machines, are implemented in the wind tunnel. A positive result in these experiments would be an important step towards proving the validity of this approach in real wind farms.

The structure of this paper will be as follows: in Section 2, the DIC strategy will be explained. Sections 3 and 4 will elaborate on the simulation environment and the experimental setup, respectively. In Section 5, the simulation results will be presented, followed by the experimental results obtained in the wind tunnel in Section 6. Finally, the conclusions will be drawn in Section 7.

**2 Control Strategy**

In this section, the strategy behind dynamic induction control will be discussed shortly. As mentioned in the introduction, the approach presented in Munters and Meyers (2018) is used as a basis for this paper: the thrust force of the upstream wind turbine is excited to induce wake mixing, in order for downstream turbines to increase their power capture. It is shown that the amplitude and frequency of a sinusoid determine the overall power production. The optimum found in here is a Strouhal number of $St = 0.25$, with an amplitude of the disk-based thrust coefficient $C_T' = 1.5$. The Strouhal number is defined as $St = fD/U_\infty$ for a given frequency $f$, rotor diameter $D$ and inflow velocity $U_\infty$, while $C_T' = 4a/(1-a)$, with $a$ the axial induction factor (Goit and Meyers, 2015). This disk-based thrust coefficient relates to the thrust coefficient $C_T$ as $C_T = C_T'(1-a)^2$. For the G1 models and an inflow velocity of $5.65$  ms$^{-1}$, this Strouhal number would result in an excitation frequency of approximately $1.3$ Hz.

However, there are some fundamental differences between Munters and Meyers (2018). First of all, due to the size of the wind tunnel (see Section 4), a 3-turbine wind farm is the deepest possible array configuration. The amplitude and frequency ranges were slightly reduced due to limits on the available time in the wind tunnel. Furthermore, the number of experiments executed in this paper is slightly lower. The amplitudes and frequencies for the wind tunnel experiments are chosen such that sufficient data points can be investigated around the optimum found in Munters and Meyers (2018). For the aero-elastic simulations, three different frequency points are evaluated to demonstrate the effect on the turbine loads. Finally, a method should be found to vary the thrust coefficient of a real (scaled) wind turbine. The thrust coefficient can be manipulated by varying either the collective pitch angle or the generator torque of the turbine. Of these two, the former approach is the most straightforward and easy to implement. Therefore, the collective pitch angle $\beta$ of the upstream model was excited periodically. This results in a slightly different thrust signal, as shown in Fig. 2, but simulations show that the difference in output for these input signals is limited. All these differences are summarized in Table 1

[Figure]

**Figure 2.** Values of $C_T$ for different types of input signals, created using a look-up table of the `G1` turbine model. The thrust coefficient is shown for three different sinusoidal excitations: on $C_T$, on $C_T'$ and on the collective pitch angle $\beta$, tuned such that the amplitude of $C_T'$ is 1.5. The dashed line shows the steady-state optimal $C_T$.

For the tests performed within the research described in this paper, the standard power controller was augmented in order to enable the rotor thrust coefficients following a specific sine wave function. However, there is not a unique way of achieving this goal, since a specific thrust coefficient $C_T(\lambda, \beta)$ can be obtained by operating at different combinations of tip-speed-ratio $\lambda$ and blade pitch $\beta$. In turn, the tip speed ratio can be varied either by changing the reference followed by the generator torque or changing the blade pitch. In this paper, a strategy that only changes the blade collective pitch is adopted. The implementation of this strategy simply requires changing the collective fine pitch at which the model blades are set when the machine operates in

**Table 1.** Differences between the approach in Munters and Meyers (2018) and both the simulations and wind tunnel experiments presented in this paper. The number of experiments executed here is slightly lower. As a result, choices are made with regards to the excitation amplitudes and frequencies that have been investigated.

|  | **Munters et al** | **Simulations** | **Experiments** |
|---|---|---|---|
| **Layout** | 4 turbines in a row | Single turbine | 3 turbines in a row |
| **Environment** | LES code | Aero-elastic code | Wind tunnel experiments |
| **Control input** | Sinusoid on $C_T'$ | Sinusoid on $\beta$ | Sinusoid on $\beta$ |
| **Amplitude of pitch excitation** | N/A | 2 | $1.7, 2.8, 5$ |
| **Amplitude of $C_T'$ excitation** | $0.5, 1, 1.5, 2$ | 1 | $1, 1.5, 2$ |
| **Number of frequency data points** | 12 | 3 | 8 |
| **Frequency range in $St$ [-]** | $[0.05, 0.6]$ | $[0.3, 0.5]$ | $[0.09, 0.41]$ |

**Table 2.** Average $\bar{C}_T$ and amplitude $A_{C_T}$ of the three different thrust coefficient oscillations whose results are discussed in Section 6, as well as the mean pitch angle average $\bar{\bar{\beta}}$ and amplitude $A_\beta$ used to achieve these signals. Note that, as explained in Section 2, these collective pitch settings are not identical for different frequencies. Instead, they are tuned such that the mean and amplitude of $C_T$ as given below are followed as accurately as possible.

| Amplitude $C'_T$ | $\bar{C}_T$ [-] | $A_{C_T}$ [-] | $\bar{\beta}$ [deg] | $A_\beta$ [deg] |
|---|---|---|---|---|
| $A = 1$ | 0.8 | 0.17 | 0.7 | 1.7 |
| $A = 1.5$ | 0.7 | 0.3 | 1.8 | 2.8 |
| $A = 2$ | 0.5 | 0.5 | 4 | 5 |

partial load conditions (region II). The fine pitch was tuned experimentally, by means of a trial and error procedure conducted with a stand-alone model, to achieving the desired mean $\bar{C}_T$ and amplitude $A$ as reported in Table 2. The effects of these control actions in terms of impacts on the power output of the 3-turbine wind farm will be discussed in Section 6.

Finally, the performance of periodic DIC as a wind farm power maximization strategy will be evaluated. To achieve this, a comparison will be made with wind farm power maximization approaches that have already been investigated more extensively in literature:

– *Greedy control*: all turbines operate at their individual optimum, disregarding wake interaction between turbines. This means that all turbine have an induction factor of $a = 1/3$ (or a thrust coefficient of $C_T = 8/9$ or $C'_T = 2$) and a yaw angle of 0 degrees with respect to the wind direction.

– *Static induction control* (also called derating control): the induction settings of upstream turbines are manipulated such that the wind farm power capture can be maximized. In this paper, the induction factor is controlled by means of the collective pitch angles of the (upstream) turbines, although using the generator torque is also an option. This strategy has been a popular research topic in recent years, and has shown both promising (Marden et al., 2013; Gebraad et al., 2013) and inconclusive (Campagnolo et al., 2016a; Nilsson et al., 2015; Annoni et al., 2016) results.

– *Yaw control* (also called wake redirection control): upstream turbines are yawed with respect to the wind direction such that the wake is steered away from downstream machines. For this approach, the control inputs are the yaw angles of the (upstream) turbines with respect to the wind. Yaw control has been demonstrated to effectively increase the wind farm power capture in wind tunnel experiments (Campagnolo et al., 2016c) and full-scale experiments (Fleming et al., 2017; Howland et al., 2019).

[revised manuscript text omitted]

Considering the class "A", where the Weibull distribution has $k = 2$ and $V_{\mathrm{av}} = 10$  ms$^{-1}$, it is possible to compute the

15 Weibull-weighted DELs for the previously considered loads. To this aim, as discussed before, DIC would normally be deactivated for wind speeds higher than 15  ms$^{-1}$. Therefore, in the second part of region III (from 17  ms$^{-1}$ to 25  ms$^{-1}$),

[Figure]

[Figure]

**Figure 9.** Comparison between tower base fore-aft bending moment (left) and hub torsional moment (right) DEL of the baseline (solid red) and DIC with $St = 0.4$ (dash-dotted blue) and $St = 0.5$ (dashed magenta) as functions of mean wind speed. The dashed yellow line indicates the rated wind velocity. Typically, DIC will only be implemented at below-rated inflow velocities.

the DELs would normally be equal to the baseline values. The Weibull-weighted DELs, computed as discussed in full operating region (from $3\,\mathrm{m/s}\,\mathrm{ms}^{-1}$ to $25\,\mathrm{m/s}\,\mathrm{ms}^{-1}$) together with the corresponding Annual Energy Production (AEP), are summarized in Table 3. As can be seen, the tower base load is affected the most (7 to $11\%$), while loads on the blade flapwise root loads increase with about $2\%$. A negligible impact is found in the blade edge-wise ($+0.4\%$) and in the hub (1 to $2\%$).

5    It is important to stress that, so far, the analyses have not considered the probability of activation of the DIC-based wind farm control, which will depend on the specific farm layout and wind rose. From this point of view, the computed DEL increments seen before, as well as the AEP decrease, are to be considered as the worst possible case, as if DIC would always be implemented regardless of wind direction and subsequent wake interaction. It is therefore possible to assess that the impact of DIC on turbine fatigue loads for the analyzed NREL 5 MW reference machine is small compared to the possible gains.

**Table 3.** Percentage increases of the Weibull-weighted DELs and AEP (from $3\,\mathrm{m/s}\,\mathrm{ms}^{-1}$ to $25\,\mathrm{m/s}\,\mathrm{ms}^{-1}$) of the excited turbine compared to the baseline for different Strouhal numbers. DIC is deactivated for wind speeds higher than $15\,\mathrm{m/s}\,\mathrm{ms}^{-1}$

|  | Blade Edgewise | Blade Flapwise | Tower ForeAft | Hub Torsion | AEP |
|---|---|---|---|---|---|
| $St = 0.3$ | +0.21% | +2.66% | + 7.06% | +0.94% | -0.46% |
| $St = 0.4$ | +0.40% | +1.80% | + 7.26% | +1.67% | -0.54% |
| $St = 0.5$ | +0.41% | +4.92% | +11.78% | +1.80% | -0.59% |

**6   Experimental Results**

In this section, the results of the experiments executed in the wind tunnel at Polimi, as described in Section 4, will be presented. The effects of periodic DIC on the power production of a 3-turbine wind farm are presented for two cases, similar to onshore and offshore wind conditions. The performance of DIC will be compared with the state-of-the-art wind farm control strategies:

5    greedy control, "static" induction control and wake redirection control.

**6.1   Power production**

First, the results with low turbulent wind (TI of approximately $5\%$) are evaluated. For this case, 3 different sets of experiments have been conducted, as defined in Table 2. These sets each represent one specific amplitude of excitation of the upstream machine: an amplitude of $A = 1$, 1.5 and 2 of $C_T'$ respectively. All other machines operate at their greedy optimum.

10    Figure 10 shows the mean power of the turbines and the total wind farm. To account for the small variations in flow conditions, the power is divided by the available power in the wind. As such, these values can be seen as power coefficients. Increasing the amplitude of the sinus decreases the power coefficient of turbine 1, while it increases the power coefficient of the downstream machines. However, for higher $A$, the loss at turbine 1 is too significant to compensate for by the downstream turbines. The unexpectedly high power loss at turbine 1 could partly be caused by a rotor imbalance that is worsened by higher

[Figure]

**Figure 10.** $\bar{C}_P$ of the wind farm in low TI conditions for different amplitudes $A$ of $C_T'$, as defined in Table 2. The bottom right figure shows the total power conversion compared to the baseline case.

**Table 4.** An overview of the total power increase with respect to the baseline case by applying dynamic induction control with different amplitudes ($A$, rows) and frequencies (columns) for the low TI case. In bold are the experiments that lead to the highest power capture for each amplitude, showing an optimum around $St = 0.28$.

[revised manuscript text omitted]

IEC 61400-1 Ed.3.: Wind Turbines — Part 1: Design requirements, Tech. rep., Garrad Hassan and Partners Ltd, St Vincent's Works, Silverthorne Lane, Bristol BS2 0QD, UK, 2004.

Munters, W. and Meyers, J.: An optimal control framework for dynamic induction control of wind farms and their interaction with the atmospheric boundary layer, Phil. Trans. R. Soc. A, 375, 20160 100, 2017.

Munters, W. and Meyers, J.: Towards practical dynamic induction control of wind farms: analysis of optimally controlled wind-farm boundary layers and sinusoidal induction control of first-row turbines, Wind Energy Science, 3, 409–425, 2018.

Nilsson, K., Ivanell, S., Hansen, K. S., Mikkelsen, R., Sørensen, J. N., Breton, S.-P., and Henningson, D.: Large-eddy simulations of the Lillgrund wind farm, Wind Energy, 18, 449–467, 2015.

Schreiber, J., Nanos, E., Campagnolo, F., and Bottasso, C. L.: Verification and calibration of a reduced order wind farm model by wind tunnel experiments, in: Journal of Physics: Conference Series, vol. 854, p. 012041, IOP Publishing, 2017.

Westergaard, C. H.: A method for improving large array wind park powre performance through active wake manipulation reducing shadow effects, 2013.